# HiSplat: Hierarchical 3D Gaussian Splatting for Generalizable Sparse-View Reconstruction

**Shengji Tang**[1,2] **Weicai Ye**[*2,3] **Peng Ye**[*2] **Weihao Lin**[1] **Yang Zhou**[1,2] **Tao Chen**[1] **Wanli Ouyang**[2]
[1]Fudan University  [2]Shanghai AI Laboratory  [3]State Key Lab of CAD&CG, Zhejiang University

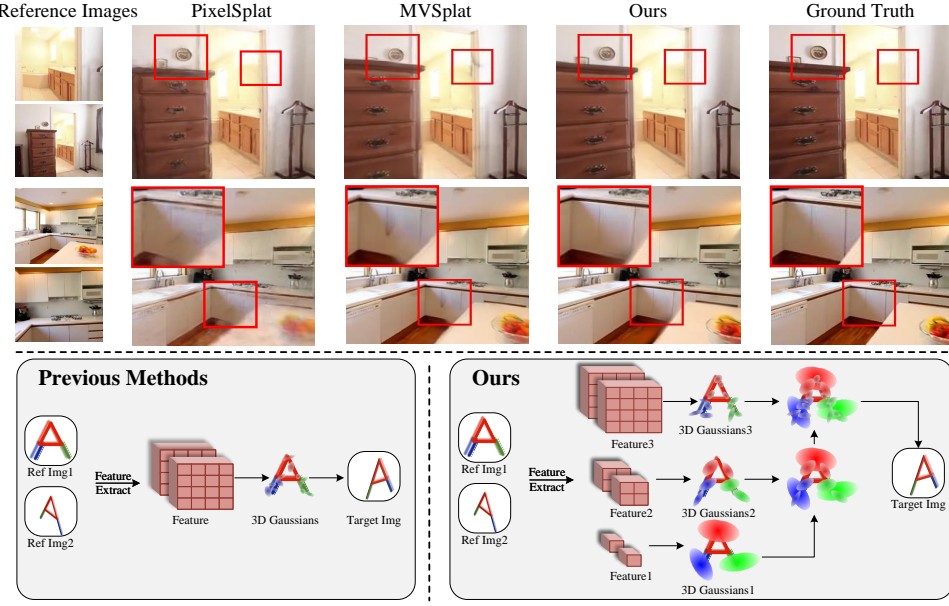

Figure 1: **Comparison between HiSplat and previous methods.** HiSplat constructs hierarchical 3D Gaussians which can better represent large-scale structures (more accurate location and less crack), and texture details (fewer artefacts and less blurriness).

## ABSTRACT

Reconstructing 3D scenes from multiple viewpoints is a fundamental task in stereo vision. Recently, advances in generalizable 3D Gaussian Splatting have enabled high-quality novel view synthesis for unseen scenes from sparse input views by feed-forward predicting per-pixel Gaussian parameters without extra optimization. However, existing methods typically generate single-scale 3D Gaussians, which lack representation of both large-scale structure and texture details, resulting in mislocation and artefacts. In this paper, we propose a novel framework, HiSplat, which introduces a hierarchical manner in generalizable 3D Gaussian Splatting to construct hierarchical 3D Gaussians via a coarse-to-fine strategy. Specifically, HiSplat generates large coarse-grained Gaussians to capture large-scale structures, followed by fine-grained Gaussians to enhance delicate texture details. To promote inter-scale interactions, we propose an Error Aware Module for Gaussian compensation and a Modulating Fusion Module for Gaussian repair. Our method achieves joint optimization of hierarchical representations, allowing for novel view synthesis using only two-view reference images. Comprehensive experiments on various datasets demonstrate that HiSplat significantly enhances reconstruction quality and cross-dataset generalization compared to prior single-scale methods. The corresponding ablation study and analysis of different-scale 3D Gaussians reveal the mechanism behind the effectiveness. Code is at https://github.com/Open3DVLab/HiSplat.

---

*Corresponding author

# 1 INTRODUCTION

As a crucial task in stereo vision, reconstructing 3D scenes from multiple viewpoints has been extensively explored. Recently, 3D Gaussian Splatting (Kerbl et al., 2023; Chen et al., 2024a) has emerged, which utilizes 3D Gaussians as explicit scene representation and adopts gradient descent to optimize the corresponding primitives. Different from previous methods based on ray-marching-based volume rendering, e.g., Neural Radiance Fields (NeRF) (Mildenhall et al., 2020), 3D Gaussian splatting (3D-GS) implements an efficient splatting rendering pipeline, enabling faster and higher-quality scene reconstruction without extra depth information. Nevertheless, the original 3D-GS requires a substantial number of multi-view images for per-scene optimization, which hinders its implementation under limited resources and sparse obversed images. To boost its generalization and transferability, data-driven generalizable 3D-GS (Szymanowicz et al., 2024; Charatan et al., 2024; Chen et al., 2024b; Zhang et al., 2024; Wewer et al., 2024) have been proposed. Generalizable 3D-GS leverages networks to feed-forward predict per-pixel Gaussian splatting parameters for unseen scenes, which can be described as mapping each 2D pixel to a fixed number of 3D Gaussians, namely splatter images. By generating splatter images, generalizable 3D-GS allows for high-quality novel view synthesis using only sparse views, especially the most challenging two views, of the new scene without additional optimization during inference.

However, current generalizable 3D-GS methods generate fixed-resolution splatter images using extracted single-scale features. The uniform 3D Gaussians result in a lack of hierarchical representation, making it challenging to simultaneously capture large-scale structures and delicate texture details. Consequently, it causes issues such as blurriness, cracks, mislocation, and artefacts as illustrated in Fig. 1. In 2D visual perception tasks such as segmentation and detection, hierarchical and multi-scale representations (He et al., 2014; Liu et al., 2015; Lin et al., 2016; Kong et al., 2018; Ghiasi et al., 2019) play an essential role in effectively capturing high-level semantic information for large-scale objects and structures while preserving low-level details. Inspired by the success of 2D multi-scale features, a natural question arises: as an explicit 3D representation akin to 2D features, **can 3D Gaussians benefit from a hierarchical structure to unleash the potential of representational capabilities?** Nonetheless, applying the hierarchical manner to generalizable 3D-GS is not straightforward. A simple preliminary experiment that extracts multi-scale features to generate hierarchical 3D Gaussians and mixes them for rendering has been conducted. As shown in the first line of Table 3, compared with the previous methods, constructing a vanilla hierarchical 3D Gaussians cannot obtain improvement. The core issue lies in that the multi-scale 3D Gaussians are generated independently and suffer from the inability to capture inter-scale information.

Different from predicting multi-scale Gaussians independently, we propose a novel hierarchical generalizable 3D-GS framework, namely **HiSplat**, for producing 3D Gaussians in a coarse-to-fine manner. Specifically, HiSplat generates large coarse-grained Gaussians to establish the large-scale structure as a skeleton, followed by fine-grained Gaussians around the coarse Gaussians to gradually enhance texture details as decoration. To prompt the interaction of different scales, we propose an attention-based **Error Aware Module** that allows finer-grained Gaussians to focus on compensating for errors generated by the coarse-grained Gaussians, referred to as Gaussian compensation. Additionally, to facilitate further correction of erroneous Gaussians, we propose a **Modulating Fusion Module** to reweight the opacities of larger-scale Gaussians based on the rendering quality of input reference images and the current joint features, termed Gaussian repair. By integrating Gaussian Compensation and Repair, the joint optimization of hierarchical 3D Gaussians can be achieved. During training, supervision is applied to the rendering images in every stage to introduce more gradient flow and provide richer depth information, enabling faster convergence and improved localization. During inference, only the final fusion Gaussians are utilized for rendering.

Extensive experiments demonstrate that, compared to previous methods with single-scale 3D Gaussian representations, HiSplat significantly enhances the quality of novel view synthesis, e.g., surpassing the leading open-source method **+0.82 PSNR** on RealEstate10K, while exhibiting stronger generalization capabilities on diverse unseen scenes, e.g., improving **+3.19 PSNR** for zero-shot testing on Replica. In summary, our contributions are as follows:

- We first study and introduce hierarchical 3D Gaussians representation in generalizable 3D-GS. It can simultaneously reconstruct higher-quality large-scale structures and more delicate texture details to significantly alleviate mislocation and artefacts.

- We construct a novel generalizable framework named HiSplat, generating hierarchical 3D Gaussians to reconstruct the scene with only two-view reference images. To exploit the inter-scale information and achieve joint optimization, we propose Error Aware Module for Gaussian compensation and Modulating Fusion Module for Gaussian repair.

- Comprehensive experiments on various mainstream datasets demonstrate that, compared with previous methods, HiSplat obtains superior reconstruction quality and cross-dataset generalization. Besides, ablation study and analysis on different scale Gaussians explain the effectiveness of HiSplat.

## 2 RELATED WORK

### 2.1 NOVEL VIEW SYNTHESIS

Novel view synthesis aims to generate photorealistic images from unseen viewpoints given a set of input images. Neural Radiance Fields (NeRF) (Mildenhall et al., 2020; Yu et al., 2021; Pumarola et al., 2020; Barron et al., 2021; 2022) marked a significant breakthrough by modeling scenes as continuous volumetric radiance fields parameterized with neural networks. Despite achieving high-quality results, NeRF suffers from slow training speeds and high memory usage due to extensive MLP evaluations and the storage of numerous point samples. In contrast, 3D Gaussian Splatting (3D-GS) (Kerbl et al., 2023; Yu et al., 2024; Yang et al., 2023) offers an explicit scene representation using anisotropic 3D Gaussians, defined by their positions, covariances, colors, and opacities. A differentiable renderer projects these Gaussians onto the image plane, allowing efficient rendering and gradient computation. However, traditional NeRF and 3DGS methods require dense multi-view images for single-scene optimization and lack generalization, performing poorly in sparse-view reconstruction tasks. Our approach addresses these limitations by achieving high generalization with only two-view reference images.

### 2.2 GENERALIZABLE 3D GAUSSIAN SPLATTING

Generalizable 3D Gaussian Splatting has become a key approach for efficient 3D scene representation and novel view synthesis. Splatter Image (Szymanowicz et al., 2024) predicts 3D Gaussian parameters from a single image using an image-to-image neural network, enabling ultra-fast single-view 3D reconstruction. PixelSplat (Charatan et al., 2024) extends this to sparse multi-view settings, using image pairs and an epipolar transformer to learn cross-view correspondences and predict depth distributions. MVSplat (Chen et al., 2024b) constructs cost volumes via plane sweeping to capture cross-view similarities, improving geometry reconstruction by predicting depth maps and unprojecting them to obtain Gaussian centers. TranSplat (Zhang et al., 2024) employs a transformer-based model for sparse-view reconstruction, using depth-aware deformable matching and monocular depth priors for refinement. Despite these advancements, existing methods often fail to capture fine-grained details due to single-scale features, leading to artefacts and reduced quality, especially in complex regions. To address these limitations, we propose a hierarchical 3D Gaussian splatting method for coarse-to-fine generalizable multi-view reconstruction.

### 2.3 HIERARCHICAL NEURAL REPRESENTATION

Hierarchical and multi-scale representations are fundamental in computer vision and graphics (Clark, 1976; Burt & Adelson, 1983; Adelson et al., 1984). In 2D visual perception, SPP-net (He et al., 2014) introduced multi-scale pooling layers for robust recognition across scales. SSD (Liu et al., 2015) utilized multi-resolution feature maps for efficient object detection, while FPN (Lin et al., 2016) constructed feature pyramids with top-down pathways and lateral connections, enabling high-level semantic features at all scales. Subsequent works (Kong et al., 2018; Ghiasi et al., 2019) continued to refine multi-scale feature representations. In 3D novel view synthesis, hierarchical approaches have enhanced efficiency and quality. NeRF (Mildenhall et al., 2020) used hierarchical sampling with coarse and fine networks. Subsequent works like KiloNeRF (Reiser et al., 2021), and PyNeRF (Turki et al., 2023) employed strategies such as spatial partitioning and multi-scale NeRF models to improve rendering speed and quality across varying scene complexities. Our work extends hierarchical principles to generalizable 3D-GS, drawing inspiration primarily from 2D vision

techniques. We are the first to apply hierarchical neural representation to explicit 3D Gaussian representations, unleashing the potential of 3D Gaussian representations in capturing rich scene details across scales.

# 3 METHOD

## 3.1 FRAMEWORK OVERVIEW

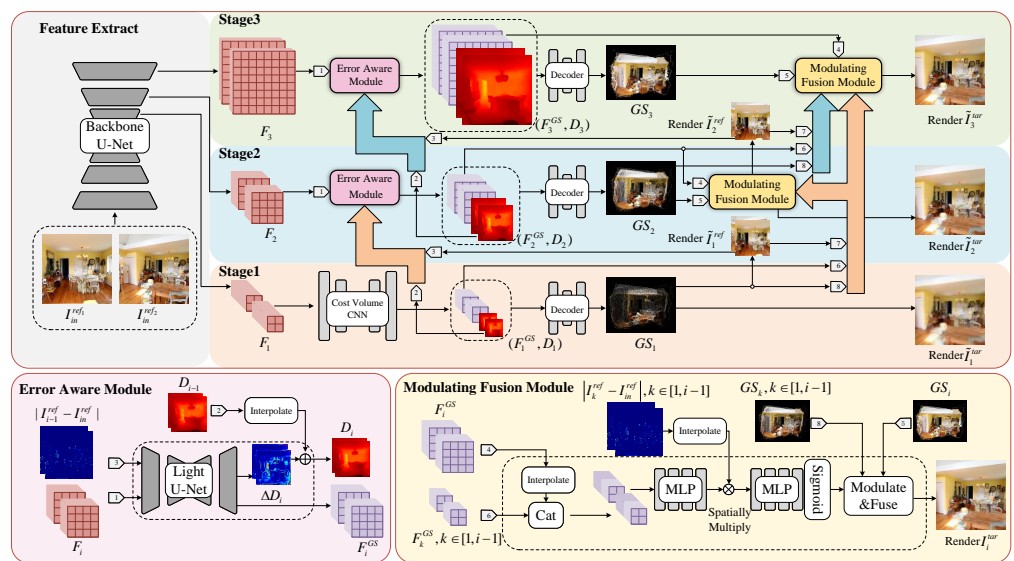

Figure 2: **The overall framework of HiSplat.** For simplicity, the situation with two input images is illustrated. HiSplat utilizes a shared U-Net backbone to extract different-scale features. With these features, three processing stages predict pixel-aligned Gaussian parameters with different scales, respectively. Error aware module and modulating fusion module perceive the errors in the early stages and guide the Gaussians in the later stages for compensation and repair. Finally, the fusing hierarchical Gaussians can reconstruct both the large-scale structure and texture details.

The overall framework of HiSplat is illustrated in Figure 2. Given a set of sparse sequential input images with their corresponding and the target camera pose information, HiSplat aims to feed-forward render the photorealistic target images from unseen views. To achieve this, HiSplat adopts a shared U-Net CNN with a multi-view transformer to extract the hierarchical cross-view features. The subsequent processing can be divided into three interrelated stages. For the lowest stage with the lowest resolution, the cross-view features are fed into a CNN with cost volume formulation to predict the depth and Gassian features, which can be decoded into corresponding Gaussian parameters. For the other two higher stages, they utilize the information (depth, features, and image errors) from the lower stages to predict their corresponding Gaussians via the Error Aware Module. Finally, the Modulating Fusion Module is used to adjust the Gaussian parameters and aggregate them to generate fusing multi-scale Gaussians. During training, fusing multi-scale Gaussians in each stage will be used to render target views parallelly with the supervision of ground truth. For inference, only the fusing Gaussians in the last stage are adopted.

## 3.2 HIERARCHICAL FEATURE EXTRACTION AND DEPTH ESTIMATION

**Hierarchical Cross-view Feature Extraction.** To extract hierarchical multi-scale features from input multi-view 2D images, we construct a CNN and Transformer mixed feature extraction backbone network with a U-Net (Ronneberger et al., 2015) architecture. Specifically, the network is divided into an encoder and a decoder, each composed of three $3 \times 3$ standard residual blocks (He et al., 2016). Each residual block performs upsampling/downsampling to extract features at different scales, and residual connections are used to integrate features of the same scale from the encoder and

decoder. At the end of the encoder, features are fed to a multi-view Transformer (Xu et al., 2023; 2022) that employs cross-attention and self-attention to extract mutual information from multi-view images and injects it into the subsequent decoding stream. For features extracted from different decoder stages, a simple convolutional head is used to obtain the corresponding scale features. To further enhance the generalization of feature extraction (see Table. 3), we follow MVSFormer++ (Cao et al., 2024) to introduce features of frozen DINOv2 (Oquab et al., 2023) and interpolate DINOv2 features to add with each scale features. Formally, Given $N$ input images $I_{in}^{ref} \in \mathbb{R}^{N \times H \times W \times 3}$ with image size $H \times W$, the cross-view feature for stage $i$ are denoted as

$$F_i = \mathcal{N}(I_{in}^{ref}; \theta_\mathcal{N})_i + Interp(\mathcal{D}(I_{in}^{ref}; \theta_\mathcal{D})), \tag{1}$$

where $F_i \in \mathbb{R}^{N \times \frac{H}{2^{3-i}} \times \frac{W}{2^{3-i}} \times \frac{C}{2^{i-1}}}$, $C$ is the channel number of $F_1$; $\mathcal{N}$ and $\mathcal{D}$ are the U-Net backbone and DINOv2 respectively; $\theta$ is the corresponding parameters of network; $\mathcal{N}(\cdot; \cdot)_i$ denotes the output features from $i_{th}$ stage of U-Net backbone decoder; $Interp(\cdot)$ is the interpolating operation.

**Depth Estimation.** With the camera matrix, depth is used to reproject the image pixel to 3D space, which is important for the correct location of Gaussians. For hierarchical Gaussians, the larger-scale Gaussians form the skeleton, while smaller Gaussians are near larger-scale Gaussians as a supplement. Therefore, the depth prediction principle is estimating the largest-scale Gaussian depth accurately, and using it as a reference to predict the depth of smaller-scale Gaussians. For depth in the first stage, we utilize the cost volume matching in Multi-View Stereo (MVS) (Cao et al., 2022; 2024; Yao et al., 2018) for accurate depth estimation. Since cost volume matching introduces unbearable computational load at high resolutions, we only apply it in the first stage. Specifically, given the feature $F \in \mathbb{R}^{N \times \frac{H}{4} \times \frac{W}{4} \times C}$ and the corresponding camera pose $P \in \mathbb{R}^{N \times 4 \times 4}$, a plane-sweep stereo approach (Collins, 1996; Yao et al., 2018; Im et al., 2019) is utilized to sample $R$ different depth candidates $V = \{d_1, d_2, ..., d_R\}$ from $[d_{near}, d_{far}]$. Then, the matching feature $F_{warp,k}^{ij} \in \mathbb{R}^{\frac{H}{4} \times \frac{W}{4} \times C}$ is sampled by warping the $j_{th}$ view feature $F^j$ to $i_{th}$ view according to depth plane $d_k$, which is denoted as

$$F_{warp,k}^{ij} = Warp(F^j, P^i, P^j, d_k) \tag{2}$$

where $Warp$ is the warping and sampling operation (Yao et al., 2018; Xu et al., 2023; Chen et al., 2024b). The matching features from other views are utilized to generate the cost volume feature $F_{cv}^i = \{F_{cv,1}^i, F_{cv,2}^i, ..., F_{cv,R}^i\}$ by matching it with $F^i$, computed as

$$F_{cv,k}^i = \frac{1}{N-1} \sum_{j=1, j \neq i}^{N} \frac{F_{warp,k}^{ij} \cdot F^i}{\sqrt{C}}. \tag{3}$$

The cost volume features are fed into a lightweight CNN (Chen et al., 2024b) to obtain refined $\hat{F}_{cv,k}^i$ and the Gaussian feature $F^{GS}$. The depth of $i_{th}$ view $D^i \in \mathbb{R}_+^{\frac{H}{4} \times \frac{W}{4}}$ is computed as

$$D^i = softmax(\hat{F}_{cv}^i)V. \tag{4}$$

For Gaussians in subsequent stages, we employ an Error Aware Module to predict their relative depth offsets, enabling the placement of Gaussians in appropriate positions where the large-scale Gaussians lack details or are incorrect.

### 3.3 ERROR AWARE MODULE

To enable small-scale Gaussians to supplement the lacking details and correct structural errors of the large-scale Gaussians, we render the mixed Gaussians from the previous stage from input views and compute an error map with the input images. A lightweight 2D U-Net (Ronneberger et al., 2015; Zhou et al., 2018b) with two predictors is used to aggregate and generate the depth offsets and Gaussian features. To ensure that the decorative Gaussians are always near the skeletal Gaussians, we have restricted the range of depth offsets. Formally, given the images $\tilde{I}_{i-1}^{ref} \in \mathbb{R}^{N \times H \times W \times 3}$ rendered from Gaussians of stage $i - 1$, the depth offsets degree $\alpha_i \in [0, 1]^{N \times \frac{H}{2^{3-i}} \times \frac{W}{2^{3-i}}}$ and Gaussian features $F_i^{GS} \in \mathbb{R}^{N \times \frac{H}{2^{3-i}} \times \frac{W}{2^{3-i}} \times C_{GS}}$ can be computed as

$$(\alpha_i, F_i^{GS}) = \mathcal{U}(|\tilde{I}_{i-1}^{ref} - I_{in}^{ref}|, F_i; \theta_\mathcal{U}), \tag{5}$$

where $\mathcal{U}$ and $\theta_{\mathcal{U}}$ are the lightweight U-Net and its corresponding weights. Given a maximum depth coefficient $\eta \in [0, 1]$ (typical value 0.1), the depth $D_i$ in stage $i$ can be computed as

$$D_i = \Delta D_i + Interp(D_{i-1}), \Delta D_i = (2\alpha_i - 1) \cdot \eta Interp(D_{i-1}), \tag{6}$$

where $Interp$ is the interpolating operation aiming to upsameple $D_{i-1}$ to the same spacial size as $D_i$. By introducing $\eta$, the $\Delta D_i$ can be restricted in $[-\eta Interp(D_{i-1}), \eta Interp(D_{i-1})]$.

### 3.4 GAUSSIAN PARAMETER PREDICTION

After obtaining the Gaussian features $F_i^{GS}$ and corresponding depth $D_i$ in stage $i$, we follow Pixel-Splat (Charatan et al., 2024) to predict the pixel-aligned Gaussian parameters, including the Gaussian center, opacity, covariance, and spherical harmonics coefficients. Specifically, for the Gaussian center, we utilize the $D_i$ along the pixel-aligned ray to unproject the 2D pixel to the 3D space location as the Gaussian center. For other Gaussian parameters, stacked convolutional layers with the corresponding predictor are adopted to generate them.

### 3.5 MODULATING FUSION MODULE

Simply fusing Gaussians of different scales struggles with achieving optimal joint optimization, and merely introducing smaller-scale Gaussians cannot fully correct the potential errors generated by large-scale Gaussians. Therefore, inspired by spatial attention (Jaderberg et al., 2015; Almahairi et al., 2016), we propose the Modulating Fusion Module. It focuses on modulating and reweighting the Gaussians' opacity in the areas with significant errors. Formally, given the Gaussian features from current and previous stages $F_i^{GS}$ and $\{F_k^{GS}|k \in [1, i-1]\}$, we obtain the concatenate Gaussians' features $F^{cat} = \{F_k^{cat}|k \in [1, i-1]\}$, denoted as

$$F_k^{cat} = cat(interp(F_i^{GS}), F_k^{GS}), \tag{7}$$

where $interp(\cdot)$ is interploating $F_i^{GS}$ to the same corresponding spatial size as each $F_k^{GS}$; $cat(\cdot, \cdot)$ is concatenating in the channel dimension. An $MLP_1$ is used to squeeze the dimension of $F^{cat}$ to multiply the corresponding error maps spatially, following another $MLP_2$ with sigmoid outputs the modulating coefficient $\xi_k \in [0, 1]^{N \times \frac{H}{2^{3-k}} \times \frac{W}{2^{3-k}}}$ of previous Gaussians' opacity $O_k \in [0, 1]^{N \times \frac{H}{2^{3-k}} \times \frac{W}{2^{3-k}}}$, denoted as

$$\xi_k = Sigmoid(MLP_2(MLP_1(F_k^{cat}; \theta_{MLP_1}) \cdot interp(|\tilde{I}_k^{ref} - I_{in}^{ref}|); \theta_{MLP_2})), \tag{8}$$

where $Sigmoid(\cdot)$ is the non-linear sigmoid operation; $\theta_{MLP}$ is the corresponding weights. The previous Gaussians' opacity $\{O_1, O_2, ..., O_{i-1}\}$ is updated following $O_k := O_k \cdot \xi_k$.

### 3.6 TRAINING OBJECTIVE

During training, we render the fusing hierarchical Gaussians in all stages to obtain images from target novel views and utilize the photometric losses (Charatan et al., 2024; Chen et al., 2024b), including Mean Squared Error (MSE) loss and Learned Perceptual Image Patch Similarity (LPIPS) losses (Zhang et al., 2018), to supervise each image. The all training objective is

$$\mathcal{L}_{all} = \sum_{i=1}^{3} \lambda_{mse}\mathcal{L}_{mse}(\tilde{I}_i^{tar}, I^{tar}) + \lambda_{lpips}\mathcal{L}_{lpips}(\tilde{I}_i^{tar}, I^{tar}), \tag{9}$$

where the loss coefficient $\lambda_{mse} = 1$, $\lambda_{lpips} = 0.05$. Except for the frozen feature extractor of DINOv2, other parameters are trainable and supervised by the training objective.

## 4 EXPERIMENT

### 4.1 EXPERIMENTAL SETTING

**Datasets.** To comprehensively evaluate the reconstruction ability, we train and test models in two large-scale datasets, RealEstate10K (Zhou et al., 2018a) and ACID (Liu et al., 2021). The

RealEstate10K dataset comprises videos sourced from YouTube, divided into 67,477 training scenes and 7,289 testing scenes. The ACID dataset consists of nature scenes captured via aerial drones, with 11,075 scenes for training and 1,972 scenes for testing. Both datasets are calibrated with Structure-from-Motion (SfM) (Schonberger & Frahm, 2016) algorithm to estimate camera intrinsic and extrinsic parameters for each frame. Following the novel view synthesis settings of previous works (Charatan et al., 2024; Chen et al., 2024b; Zhang et al., 2024), two context images are as input, and three novel target views are rendered for each test scene. Besides, to compare the cross-dataset generalization ability, we select other two multi-view datasets, including real object-centric dataset DTU (Jensen et al., 2014) and synthetic indoor dataset Replica (Straub et al., 2019), for zero-shot test (without fine-tuning or training). Following (Chen et al., 2024b) and (Zhi et al., 2021), we select sixteen scenes in DTU and eight scenes in Replica for testing. To quantitatively measure the rendering quality of novel views from different aspects, Peak Signal-to-Noise Ratio (PSNR), Structural Similarity (SSIM) (Wang et al., 2004) and Learned Perceptual Image Patch Similarity (LPIPS) losses (Zhang et al., 2018) are adopted as testing metrics.

**Implementation Details.** For a fair comparison, we follow the commonly used training settings (Chen et al., 2024b; Charatan et al., 2024; Zhang et al., 2024). Specifically, the input images are resized as $256 \times 256$, and the model is optimized by Adam (Kingma, 2014) for 300,000 iterations. The training experiments are implemented in 8 RTX4090 with batch size 2 for two days. There are more implementation details in A.1.

## 4.2 MAIN RESULTS

### 4.2.1 NOVEL VIEWS SYNTHESIS

To verify the effectiveness of the proposed HiSplat on novel view synthesis, we train and test the model on large-scale multi-view datasets RealEstate10K and ACID, respectively. We select four typical feed-forward NeRF-based methods, including pixelNeRF (Yu et al., 2021), GPNR (Suhail et al., 2022), AttnRend (Du et al., 2023), MuRF (Xu et al., 2024), and three recent Gaussian-Splatting-based methods, including PixelSplat (Charatan et al., 2024), MVSplat (Chen et al., 2024b), TranSplat (Zhang et al., 2024) as comparisons. As shown in Table 1, compared with previous methods, HiSplat can consistently achieve state-of-the-art (SOTA) performance on different datasets and metrics with significant improvement, which demonstrates the superiority of the proposed hierarchical structure of Gaussian primitives. Specifically, on RealEstate10K dataset, HiSplat surpasses the latest state-of-the-art (SOTA) open-source method MVSplat by +0.82 PSNR, and exceeds the leading method Transplat by +0.52 PSNR, achieving a new milestone by obtaining higher than 27 PSNR in the challenging two-view reconstruction task. On ACID dataset, HiSplat outperforms other methods by +0.5 PSNR than MVSpalt and +0.4 PSNR than TranSplat. For the patch-level SSIM and feature-level LPIPS metrics, HiSplat also gains significant improvement, suggesting that HiSplat can reconstruct details and large-scale structures with higher quality. Besides the performance, we also report the inference time and peak GPU memory in A.2.

Table 1: **Evaluation on RealEstate10K and ACID.** Compared with the previous NeRF-based and generalizable Gaussian-Splatting-based method, HiSplat can consistently obtain higher rendering quality of unseen views.

| Method | RealEstate10K | | | ACID | | |
|---|---|---|---|---|---|---|
| | PSNR ↑ | SSIM ↑ | LPIPS ↓ | PSNR ↑ | SSIM ↑ | LPIPS ↓ |
| pixelNeRF (Yu et al., 2021) | 20.43 | 0.589 | 0.550 | 20.97 | 0.547 | 0.533 |
| GPNR (Suhail et al., 2022) | 24.11 | 0.793 | 0.255 | 25.28 | 0.764 | 0.332 |
| AttnRend (Du et al., 2023) | 24.78 | 0.820 | 0.213 | 26.88 | 0.799 | 0.218 |
| MuRF (Xu et al., 2024) | 26.10 | 0.858 | 0.143 | 28.09 | 0.841 | 0.155 |
| PixelSplat (Charatan et al., 2024) | 25.89 | 0.858 | 0.142 | 28.14 | 0.839 | 0.150 |
| MVSplat (Chen et al., 2024b) | 26.39 | 0.869 | 0.128 | 28.25 | 0.843 | 0.144 |
| TranSplat (Zhang et al., 2024) | 26.69 | 0.875 | 0.125 | 28.35 | 0.845 | 0.143 |
| HiSplat(Ours) | 27.21 | 0.881 | 0.117 | 28.75 | 0.853 | 0.133 |

### 4.2.2 CROSS-DATASET GENERALIZATION

To verify the generalization ability of HiSplat, we train the model on RealEstate10K and directly test it on DTU, ACID, and Replica in a zero-shot setting. As depicted in Figure 3, the images generated by HiSplat are more aligned with human perception, featuring less edge blurriness, less artefacts

and more accurate location. The quantitive results are shown in Table 2. Compared with previous single-scale Gaussian-Splatting-based methods, HiSplat performs a significant generalization improvement, e.g., obtaining +1.05 PNSR on object-centric dataset DTU, +0.55 PSNR on outdoor dataset ACID and +3.19 PSNR on indoor dataset Replica over the suboptimal methods, suggesting that HiSplat is more effectively deployed in the practical open-world scenario with various data distribution. It is worth noting that on ACID, HiSplat's zero-shot performance even outperforms others trained specially on ACID significantly. The outstanding generalization ability can be explained from two aspects: 1) the hierarchical Gaussian representation can reconstruct scenes of vastly different scales simultaneously 2) the error-aware mechanism is scale-invariant, which can benefit consistently across a wide range of scenes.

Table 2: **Evaluation on cross-dataset generalization.** We train models on RealEstate10K, and test them on object-centric dataset DTU, outdoor dataset ACID, and indoor dataset Replica in a zero-shot setting. Compared with previous methods, HiSplat can better handle various scenes with different distributions and scales.

| Method | RealEstate10K → DTU | | | RealEstate10K→ACID | | | RealEstate10K→Replica | | |
|---|---|---|---|---|---|---|---|---|---|
| | PSNR ↑ | SSIM ↑ | LPIPS ↓ | PSNR ↑ | SSIM ↑ | LPIPS ↓ | PSNR ↑ | SSIM ↑ | LPIPS ↓ |
| PixelSplat (Charatan et al., 2024) | 12.89 | 0.382 | 0.560 | 27.64 | 0.830 | 0.160 | 23.98 | 0.821 | 0.202 |
| MVSplat (Chen et al., 2024b) | 13.94 | 0.473 | 0.385 | 28.15 | 0.841 | 0.147 | 23.79 | 0.817 | 0.165 |
| TranSplat (Zhang et al., 2024) | 14.93 | 0.531 | 0.326 | 28.17 | 0.842 | 0.146 | - | - | - |
| HiSplat(Ours) | 16.05 | 0.671 | 0.277 | 28.66 | 0.850 | 0.137 | 27.17 | 0.899 | 0.113 |

## 4.3 ABLATION EXPERIMENT

To verify the effectiveness of each proposed technique of HiSplat, we conduct ablation experiments to eliminate each component step by step, and compare them with previous methods on RealEstate10K. As illustrated in Table 3, different degrees of performance degradation are observed of any removal, validating the necessity of each component. It merits attention that the vanilla hierarchical 3D Gaussian representation cannot perform competitively compared with previous methods (-0.21 PSNR compared with MVSpalt). Whereas, when combined with the proposed EAM and MFM, the potential of hierarchical manner is unleashed with prominent improvement (+0.82 PSNR). The reason is that the EAM and MFM promote the transfer of error-aware information and features across different scales, driving Gaussians in the later stages to compensate for lacking details and repair the error of earlier stages for joint optimization, as stated in Sec 1.

Table 3: **The ablation study on RealEstate10K.** Hier: Hierarchical Gaussian, EAM: Error Aware Module, MFM: Modulating Fusion Module, DINO: using DINOv2 feature. Only using vanilla hierarchical Gaussians cannot perform better than previous methods. Each proposed component contributes to the final superior results.

| | Hier | EAM | MFM | DINO | PSNR↑ | SSIM↑ | LPIPS↓ |
|---|---|---|---|---|---|---|---|
| Variants of HiSplat(Ours) | ✓ | × | × | × | 26.18 | 0.869 | 0.135 |
| | ✓ | ✓ | × | × | 26.76 | 0.874 | 0.123 |
| | ✓ | ✓ | ✓ | × | 27.02 | 0.879 | 0.120 |
| | ✓ | ✓ | ✓ | ✓ | 27.21 | 0.881 | 0.117 |
| Other methods | PixelSplat (Charatan et al., 2024) | | | | 25.89 | 0.858 | 0.142 |
| | MVSplat (Chen et al., 2024b) | | | | 26.39 | 0.869 | 0.128 |

## 4.4 ANALYSIS OF HIERARCHICAL GAUSSIANS

In this section, we visually analyze the different stages during the inference process to reveal the mechanism behind the effectiveness of HiSplat.

**Analysis of 3D Gassian Primitives.** To visually display the fusing Gaussians in different stages, we take the center of Gaussian primitives as the 3D position of point clouds and utilize the color of corresponding context images to draw the Gaussian primitives. As shown in Figure 4, for the later stage, as the number of Gaussian primitives increases, the rendering quality also gradually improves, demonstrating the effectiveness of adding Gaussian primitives. To further analyze the function of adding Gaussians of each stage, we report the statistical probability of opacity and mean scale of Gaussians. It is noteworthy that because the fusing Gaussians in the later stage contain the Gaussians from the early stages, for a clear analysis, we individually report the statistical value of fusing Gaussians from each stage. It can be observed that the Gaussian primitives in stage 1 are sparse, solid,

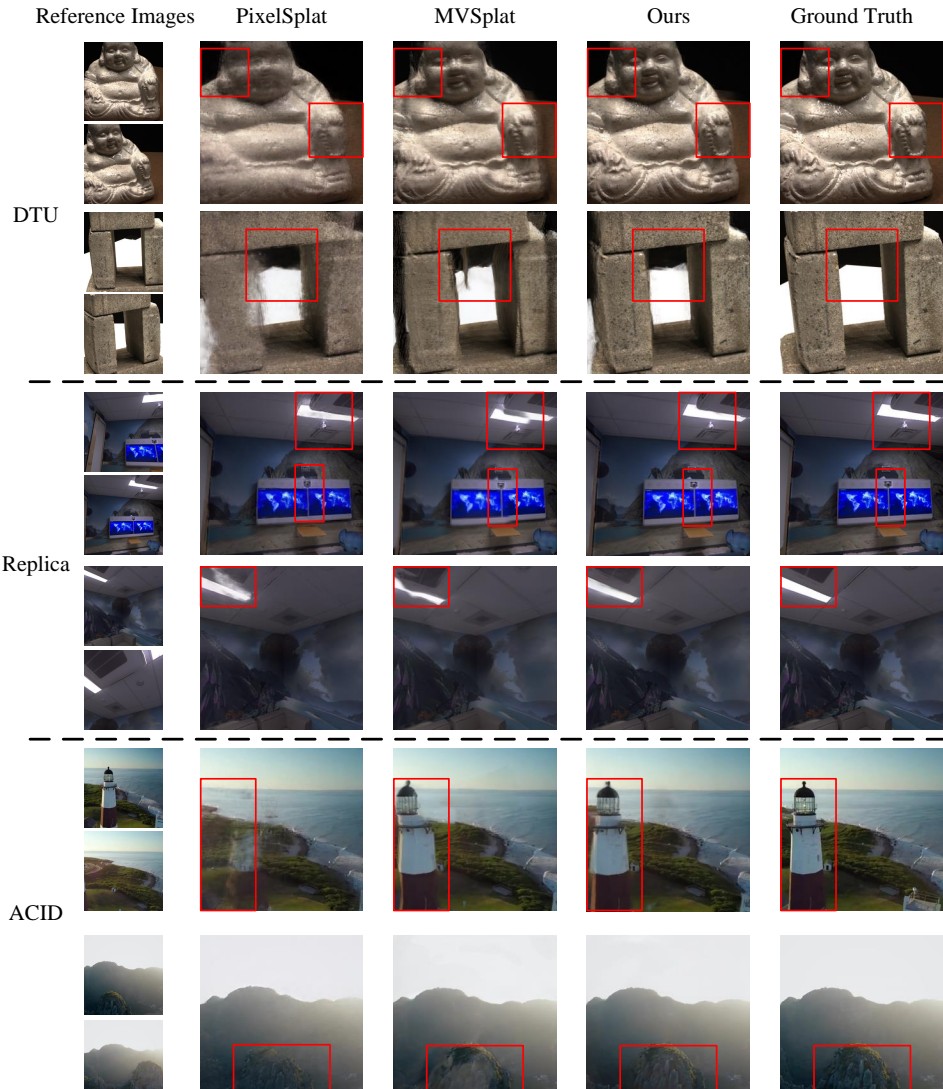

Figure 3: **Qualitative comparison of generalization ability.** For the scenes out of training distribution, HiSplat can generate higher-quality novel-view images. More comparison is provided in A.3.

and large while the adding Gaussian primitives from later stages gradually become dense, transparent and small. This pattern of Gaussian attributes aligns with our hypothesis: **larger and more solid Gaussian primitives form the skeleton of the scene, aiming to establish the large-scale basic structure, while smaller and more transparent Gaussian primitives serve as decoration, further refining the texture details, which can be described as "from bone to flesh".** Following this hypothesis, we explain the function of the displayed hierarchical Gaussians that Gaussian primitives from stage 1 and stage 2 are relatively solid, adhering closely to the surface of the scene, which gradually completes the outline and basic structure; as for the Gaussian primitives from stage 3, they are very tiny and transparent, introducing richer texture details and illumination. Besides, compared with other methods, HiSplat can render a high-quality novel-view image with more accurate and consistent geometry, especially the toy bird's crest in Figure 4.

**Analysis of 2D Error Map.** To analyze how hierarchical Gaussians reconstruct the scene step by step in detail, we show the colored error map of different stages, generated by comparing the rendering images with the ground truth in Figure 5. The brighter and redder pixels exhibit more significant errors and a rectangle highlights the regions with complex and rich textures. It can be observed that as the stage index increases, the overall error continuously decreases, especially in areas with complex textures. Besides, by directly examining the rendered images, we can observe

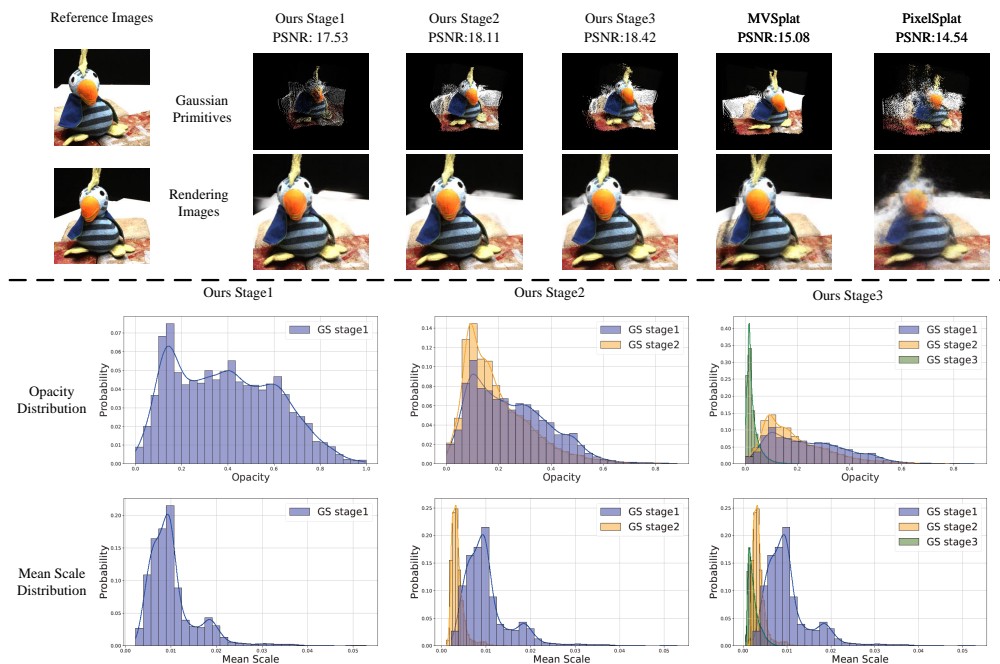

Figure 4: **Comparison of Gaussian primitives in different stages on DTU.** HisPlat can gradually generate large-scale solid Gaussians as "bone" and small-scale transparent Gaussians as "flesh", confirming better rendering quality and geometry.

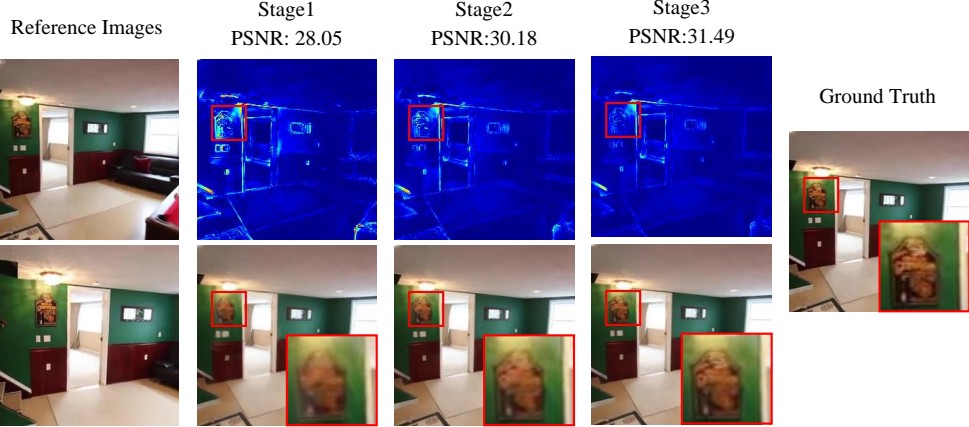

Figure 5: **Comparison of rendering images from different stages on RealEstate10K.** HiSplat can perceive the error, and utilize Gaussians in the later stages to add details and correct errors gradually.

that there is less blurriness and more details. It suggests that HiSplat can perceive the error in the early stage, and utilize Gaussians in the later stage for repair and compensation.

## 5 CONCLUSION

In this paper, we propose a novel generalizable 3D Gaussian Splatting framework, HiSplat, aiming to render novel-view images from sparse (only two) context images. Different from previous methods predicting single-scale Gaussians, HiSplat gradually generates multi-scale hierarchical Gaussians to reconstruct the large-scale structure and texture details with higher quality. To facilitate the information interaction of different stages, we propose Error Aware Module and Modulating Fusion Module for Gaussian compensation and repair. Extensive experiments across multiple datasets demonstrate that HiSplat significantly enhances the quality of reconstructions and cross-dataset generalization, surpassing previous single-scale methods.

## ACKNOWLEDGEMENTS

This work is supported by National Key Research and Development Program of China (No. 2022ZD0160101), Shanghai Natural Science Foundation (No. 23ZR1402900). The computations in this research were performed using the CFFF platform of Fudan University.

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
