# HiSplat: Hierarchical 3D Gaussian Splatting for Generalizable Sparse-View Reconstruction

**Shengji Tang**[1,2] **Weicai Ye**[*2,3] **Peng Ye**[*2] **Weihao Lin**[1] **Yang Zhou**[1,2] **Tao Chen**[1] **Wanli Ouyang**[2]
[1]Fudan University    [2]Shanghai AI Laboratory    [3]State Key Lab of CAD&CG, Zhejiang University

## A  Appendix

### A.1  Detailed architecture and experiment setting

In this section, we provide more details of the architecture and experiments.

**More Architecture details.** For the U-Net backbone, the encoder and decoder are constructed with a similar architecture consisting of stacked residual blocks. The kernel size of each block is $3 \times 3$, with increasing channels $[32, 64, 128]$ and half resolution for each stage. For combining the U-Net features with DINOv2 features, we follow the Side View Attention of MVSformer++ (Cao et al., 2024) to further improve the information aggregation from different views. There is a convolution layer to align the channels of DINOv2 features with the corresponding U-Net features for addition. Then, another convolutional layer is used to generate the features of each stage.

**More Experiment details.** For the training schedule on ACID and RealEstate10K, we follow (Chen et al., 2024) to set a learning rate of 0.0002 with warm-up cosine decay. The warm-up steps are 2000. Random horizontal flipping is used as data augmentation for each image set. All experiments are based on the deep learning framework PyTorch and utilize Pytorch-Lightning for better organization. For results on RealEstate10K and ACID, because our basic experiment setting is as same as (Chen et al., 2024; Charatan et al., 2024; Zhang et al., 2024), we directly cite the results from the original literature. For the zero-shot results on Replica, we only conduct experiments to compare with open-source methods MVSplat and PixelSplat, using their latest official checkpoints.

### A.2  Efficiency and Effectiveness of different stages

To report the effectiveness and efficiency of the proposed HiSplat, we test the inference cost and performance of HiSplat and previous methods. All experiments are conducted on an RTX4090 with batch size 8. The input image size is $256 \times 256$, and the reported inference time and peak memory are the average value of 10,000 recorded data points with the same inference settings. Because the early-stage Gaussians are predicted without relying on the later-stage Gaussians, HiSplat can terminate the inference in the early stages and reduce the inference cost, we add two variants of HiSplat, HiSplat-Stage 1 and HiSplat-Stage 2, which only predict the Gaussians in the stage 1 and 2, respectively. Thanks to the hierarchical character of HiSplat, it is very simple to change the mode of HiSplat for a flexible implementation. The results are shown in Table 1. It can be observed that compared with the previous methods, the complete version of HiSplat (HiSplat-Stage 3) can obtain the best performance (+0.82 PSNR compared with MVSplat) with relatively high but bearable inference cost, due to the three-stage processing and the introduction of DINOv2 feature. Flash Attention (Dao, 2023; Dao et al., 2022) and stage-wise parallel techniques are the optimal methods for future cost reduction. Besides, other modes of HiSplat can properly sacrifice the performance for lightweight implementation. Thus, compared with the previous fixed methods, HiSplat has the potential to form a dynamic framework for various practical requirements of resource or quality, suggesting HiSplat can achieve a better balance between performance and inference cost.

---

[*]Corresponding author

Table 1: **The efficiency and effectiveness of HiSplat and other methods.** HiSplat-Stage 1,2 represents only forwarding the 1,2 stages of HiSplat to generate part of hierarchical Gaussians for rendering, which can sacrifice performance for higher efficiency. Under different resource constraints, HiSplat can switch inference mode flexibly, achieving a good balance between performance and inference cost.

| Method | Peak Memory/GB↓ | Inference Time/s↓ | PSNR↑ | SSIM↑ | lPIPS↓ |
|---|---|---|---|---|---|
| PixelSplat (Charatan et al., 2024) | 24.17 | 1.47 | 25.89 | 0.858 | 0.142 |
| MVSplat (Chen et al., 2024) | 14.08 | 0.27 | 26.39 | 0.869 | 0.128 |
| HiSplat-Stage 1 | 12.91 | 0.24 | 26.13 | 0.858 | 0.141 |
| HiSplat-Stage 2 | 14.74 | 0.36 | 26.99 | 0.879 | 0.120 |
| HiSplat-Stage 3 | 23.34 | 0.51 | 27.21 | 0.881 | 0.117 |

## A.3 MORE QUALITATIVE COMPARISON

For a more detailed visual comparison, there are extended qualitative comparisons including novel-view-synthesis results on RealEstate10K in Figure 1; zero-shot results on DTU in Figure 2, ACID in Figure 3, Replica in Figure 4.

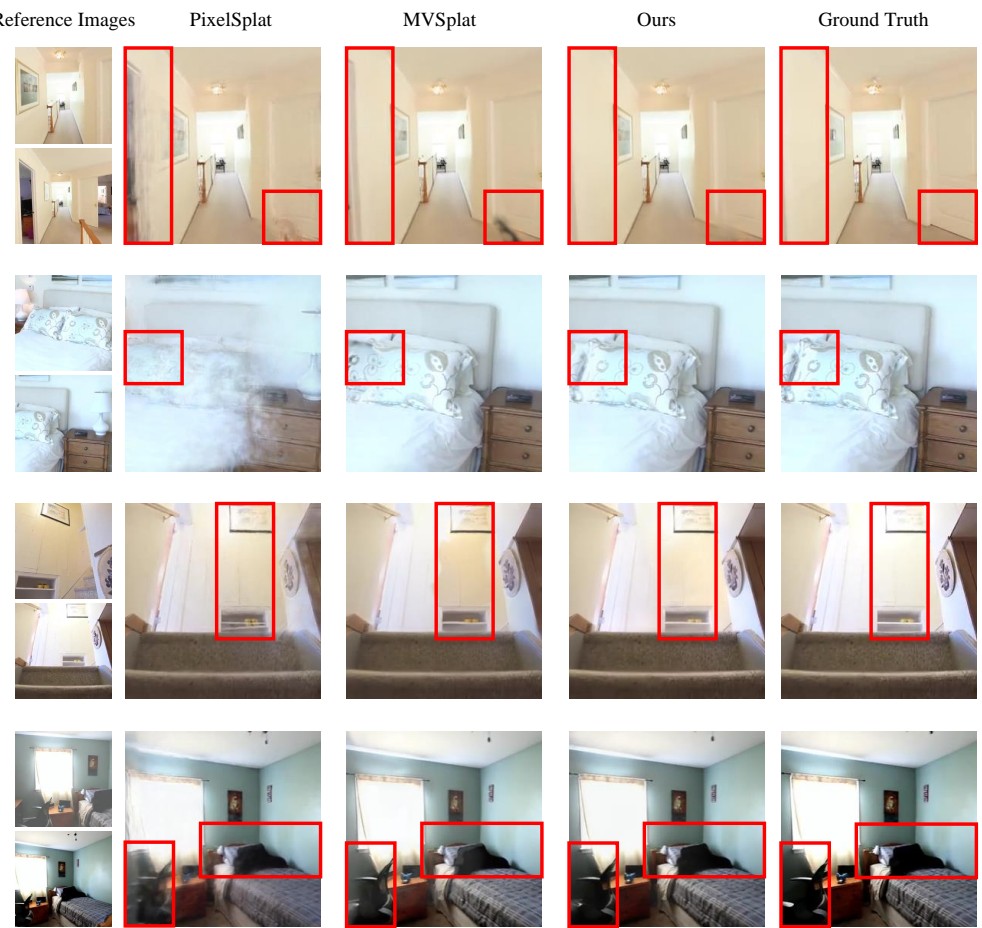

Figure 1: **Extended qualitative comparison on RealEstate10K.** Compared with the previous methods, HiSplat can handle large-scale structures (the location or shape of objects) and texture details better.

| Reference Images | PixelSplat | MVSplat | Ours | Ground Truth |
|---|---|---|---|---|

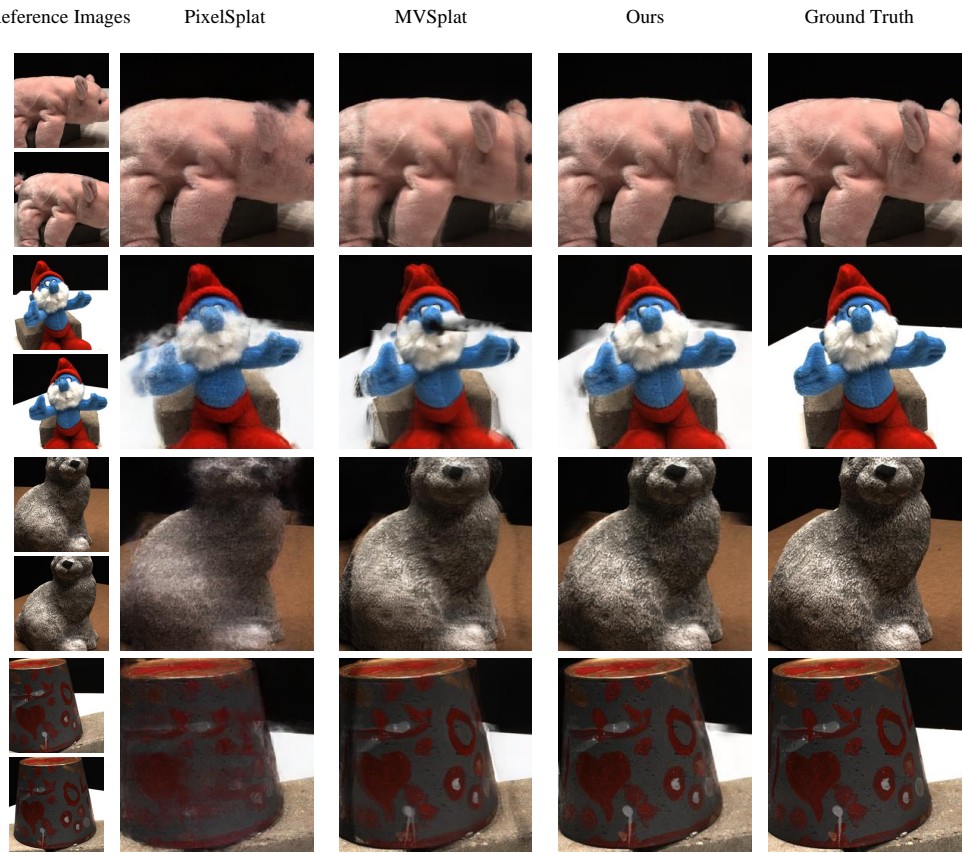

Figure 2: **Extended qualitative comparison on DTU.** For unseen objects, HiSplat can reconstruct the texture details more clearly with fewer artefacts and less blurriness.

## A.4 POSE-FREE SPARSE-VIEW RECONSTRUCTION

By integrating DUSt3R (Wang et al., 2024a), HiSplat can easily perform sparse-view reconstruction without relying on pre-calibrated camera parameters. To evaluate the HiSplat's ability to handle pose-free scenarios, we conduct a simple experimental setup to test it on the DTU and Replica datasets, using sparse views without camera parameters. Specifically, we use pre-trained DUSt3R to extract camera parameters, including intrinsics and extrinsics, from input images and then directly apply HiSplat (pre-trained on RealEstate10K) for novel view synthesis. The results are shown in Table 2. Compared with using relatively accurate poses from the dataset, using poses generated by DUSt3R leads to reduced performance but still outperforms MVSplat. This is because the simple experimental setup lacks sufficient coupling between DUSt3R's pose extraction and HiSplat's reconstruction process. The resulting less accurate poses affect HiSplat's ability to precisely determine the position and shape of Gaussian primitives, thus leading to reduced performance.

Additionally, we test in real-life scenarios using three casually taken photos with a smartphone: two as reference images and one as the ground truth. Using the same simplified pipeline described above, the results are shown in Fig. 5. It demonstrates that HiSplat+DUSt3R's simplified pipeline has the ability to handle sparse-view reconstructions in real-life settings where only the images are inputted, producing a slightly satisfactory novel view synthesis. We believe designing a more robust and systematic pose-free reconstruction method based on HiSplat is a promising future direction.

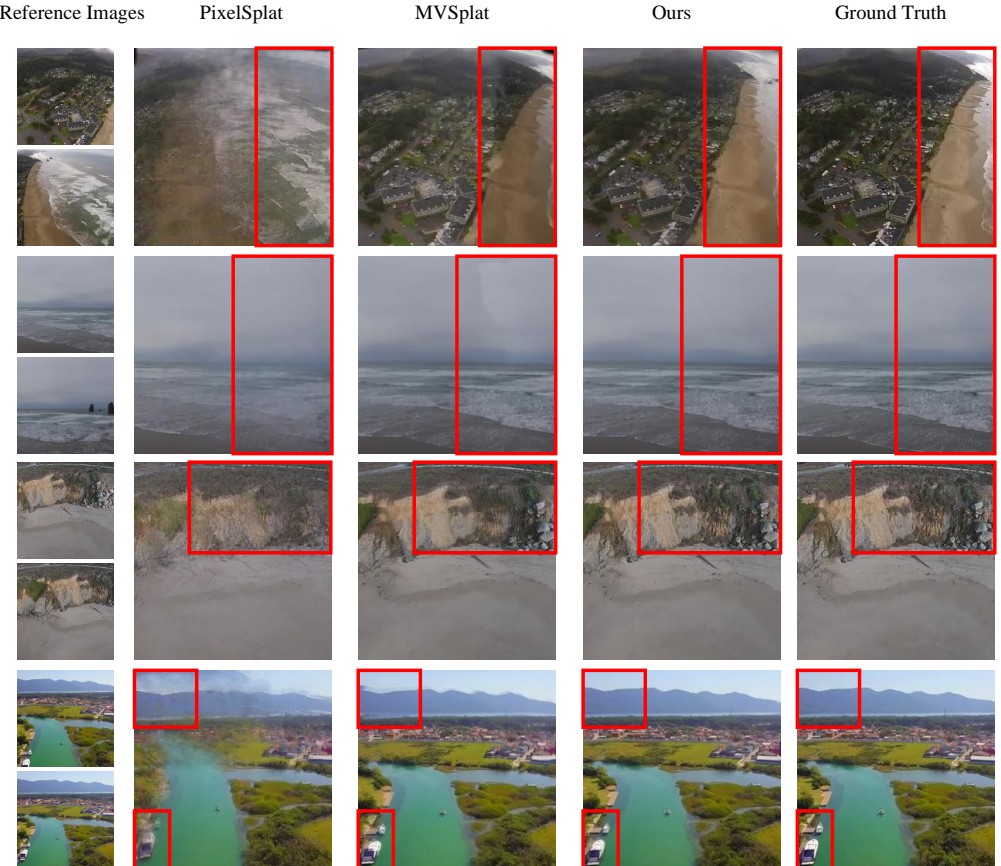

Figure 3: **Extended qualitative comparison on ACID.** For unseen outdoor coastal scenes, HiSplat can generate more accurate illumination and location with fewer artefacts.

Table 2: **Cross-dataset evaluation with pose-free settings.** HiSplat+GT pose: using the camera parameters from datasets; HiSplat+DUSt3R pose: using the camera parameters generated by pretrained DUSt3R, which is used for pose-free settings. By simply combining with DUSt3R, HiSplat can handle pose-free sparse-view reconstruction.

| Method | RealEstate10K→DTU | RealEstate10K→Replica |
|---|---|---|
| MVSplat+GT pose | 13.94 | 23.79 |
| MVSplat+DUSt3R pose | 13.18 | 21.04 |
| HiSplat+GT pose | 16.05 | 27.21 |
| HiSplat+DUSt3R pose | 14.58 | 22.20 |

### A.5 Implementation for more reference views

Although HiSplat focuses on two-view reconstruction, the main components of HiSplat, such as the hierarchical Gaussian representation, shared feature extractors, Error Aware Module, and Modulating Fusion Module, are view-invariant, which means that HiSplat can be easily extended to handle reconstruction from an arbitrary number of views. To perform a preliminary verification, we directly use the weights trained by inputting two views on RealEstate10K to conduct cross-dataset testing with three views on DTU and Replica. The results are shown in Table 3. Compared with the two-view case, the performance slightly improves with inputting three views, and HiSplat still outperforms MVSplat, demonstrating the feasibility of using HiSplat for multi-view reconstruction directly. However, since HiSplat is not specifically trained for three-view tasks and lacks specific structural adjustments for more views in Table 3, the gain is marginal.

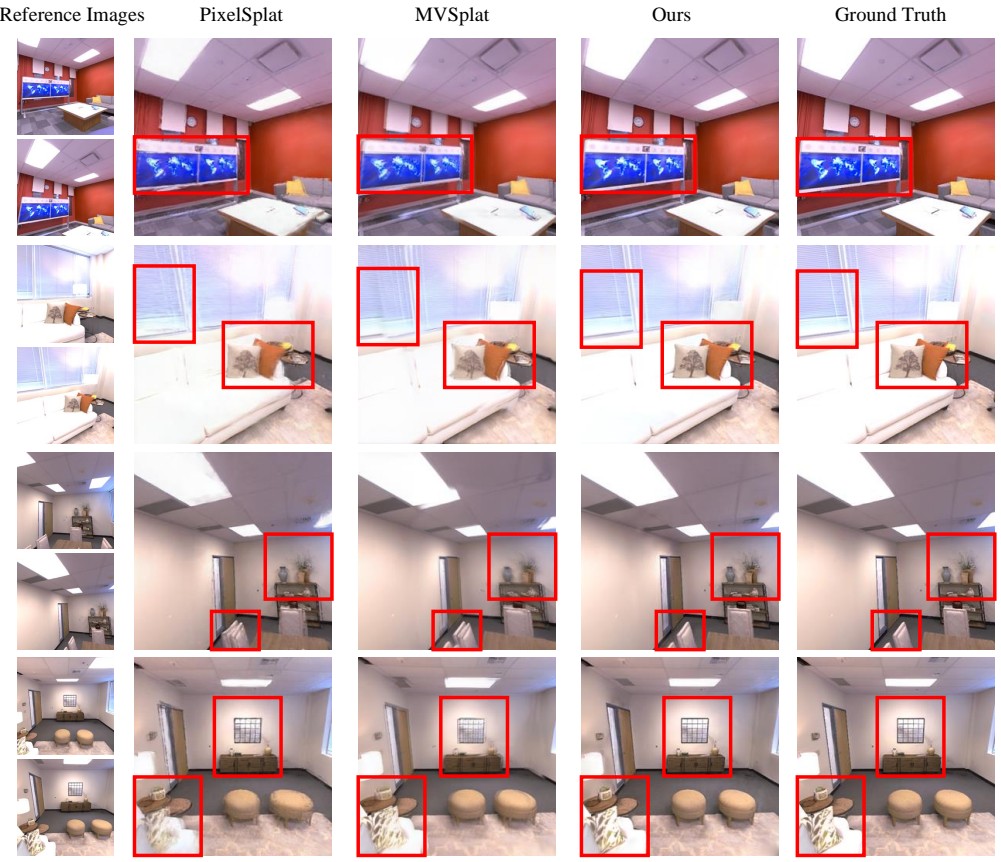

Figure 4: **Extended qualitative comparison on Replica.** For unseen indoor scenes, HiSplat can reconstruct the details better with less blurriness and fewer artefacts.

Table 3: **Cross-dataset evaluation with more reference views.** We train the models on RealEstate10K with two reference views and test them on DTU and Replica with three reference views.

| Method | RealEstate10K→DTU | RealEstate10K→Replica |
|---|---|---|
| MVSplat+2 views | 13.94 | 23.79 |
| MVSplat+3 views | 14.29 | 24.75 |
| HiSplat+2 views | 16.05 | 27.17 |
| HiSplat+3 views | 16.35 | 27.47 |

A.6 COMPARISON OF DIFFERENT GAUSSIAN PRIMITIVE NUMBER

In this section, we discuss the impact of the number of Gaussian primitives. In the third stage, the number of Gaussian primitives is the same as in MVSplat. However, as the reduction of stage index is reduced, the number of Gaussian primitives decreases exponentially by a factor of 4, resulting in significantly fewer Gaussian primitives in the early stages. Therefore, the total number of Gaussian primitives is only 1.3125 (0.0625 + 0.25 + 1) times compared with MVSplat. The comparison of the number of Gaussian primitives and performance is shown in Table 4. Compared with other methods, HiSPlat achieves significant performance improvement with only a slight increase in Gaussian primitives. It is worth noting that even if MVSplat triples the number of Gaussian primitives, it does not yield noticeable improvement, indicating that the increase in Gaussian primitives is not the main reason for HiSplat's performance gain.

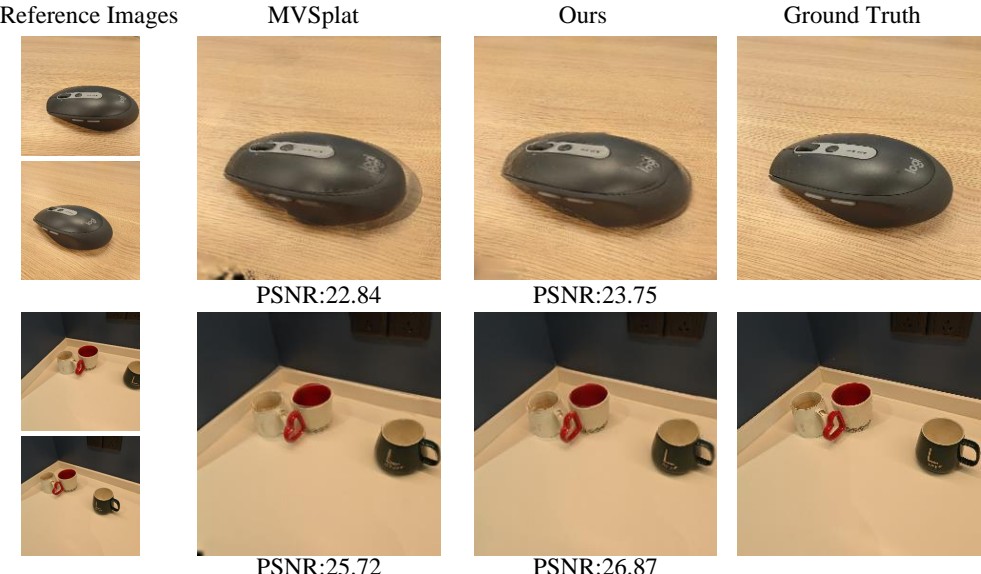

Figure 5: **Pose-free sparse-view reconstruction in real-life scenarios.** We take three photos using a smartphone: two as reference images and one as the ground truth. Then, a pre-trained DUSt3R is used to extract the camera parameters from sparse views. The generated camera parameters and two reference images are inputted in MVSplat or HiSplat for novel view synthesis.

Table 4: **Comparison of different Gaussian primitive number.** Compared with other pixel-wise-Gaussian-based methods, HiSplat utilizes hierarchical Gaussian representation, which increases marginally the number of Gaussian primitives and boosts the performance to achieve a better trade-off.

| Method | Gaussian Primitive Number | PSNR↑ | SSIM↑ | LPIPS↓ |
|---|---|---|---|---|
| PixelSplat | 65,632(×1.0) | 25.89 | 0.858 | 0.142 |
| MVSplat(1 Gaussian per pixel) | 65,632(×1.0) | 26.39 | 0.869 | 0.128 |
| MVSplat(3 Gaussian per pixel) | 196,896(×3.0) | 26.54 | 0.872 | 0.127 |
| HiSplat | 86,142(×1.3125) | 27.21 | 0.881 | 0.117 |

## A.7 MORE GENERALIZATION CASES

To better display the generalization capability of HiSplat, we provide additional visualization cases generated by HiSplat trained on Re10K and tested on DTU. Refer to Fig. 6 for details.

## A.8 MORE RELATED WORKS

In this section, we supplement more related works. FreeSplat (Wang et al., 2024b)proposed a feed-forward generalizable model tailored for indoor scenes, capable of reconstructing global 3D Gaussians from arbitrary numbers of input views. It introduces efficient cross-view feature aggregation and a pixel-wise triplet fusion module to reduce Gaussian redundancy and aggregate multi-view features. Although FreeSplat also employs a multi-scale U-Net as an efficient feature aggregator, the features it ultimately uses to predict Gaussian attributes are from the last layer and still single-scale, resulting in single-scale Gaussian primitives without hierarchical structure. On the contrary, HiSplat leverages multi-scale features from different layers of the U-Net to generate different-scale Gaussians, which represent the scene from bone to flesh.

Splatt3R (Smart et al., 2024) introduced a pose-free, feed-forward method for 3D reconstruction and novel view synthesis from uncalibrated image pairs. Building upon DUSt3R's architecture (Wang et al., 2024a), it predicts 3D Gaussian primitives without requiring camera parameters or depth

Reference Images       PixelSplat       MVSplat       Ours       Ground Truth

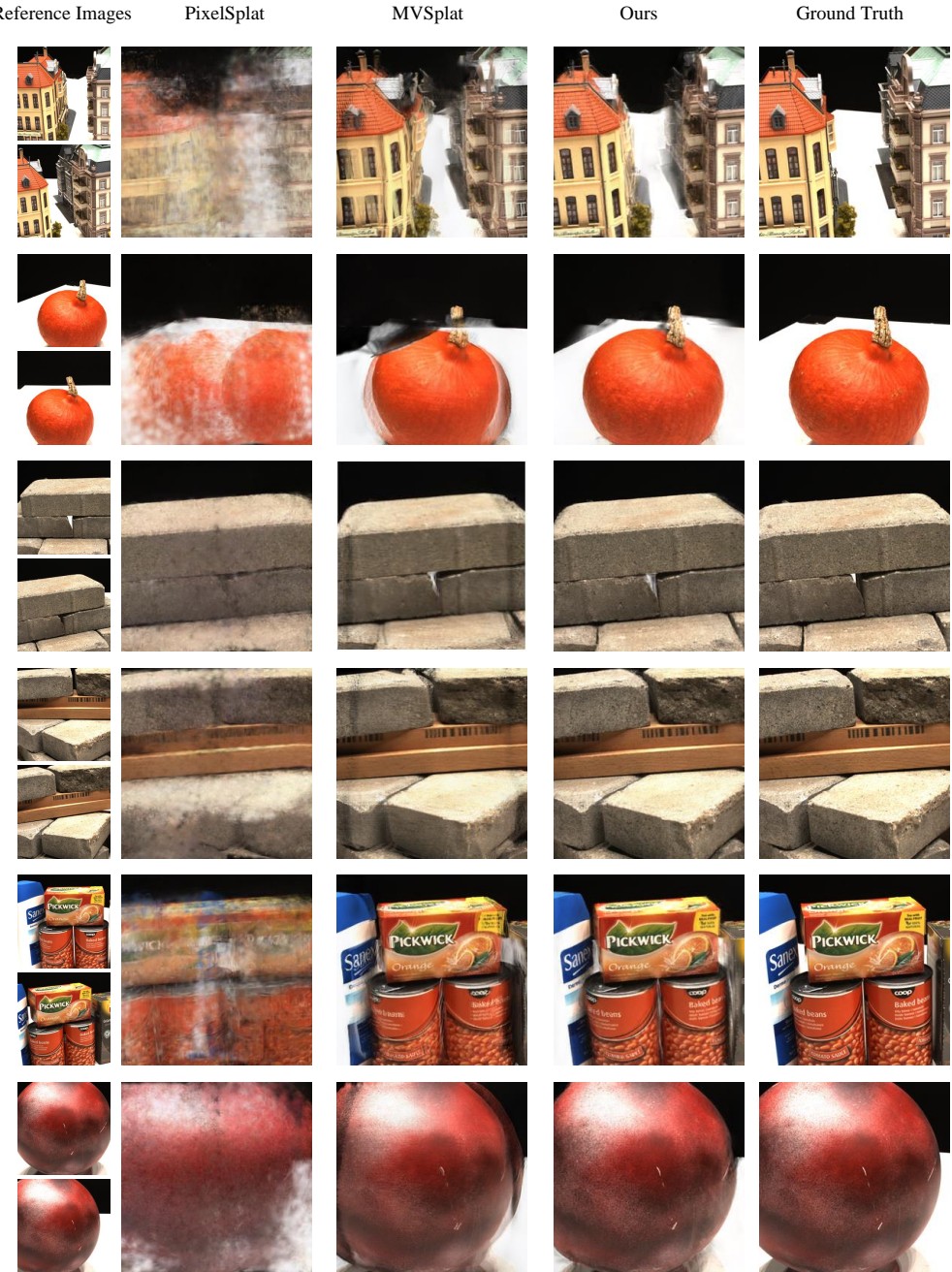

Figure 6: **More cases on DTU to demonstrate generalization ability.**.

information. As a promising direction for improvement, HiSplat has the potential to be combined with Splatt3R to handle pose-free scenarios.

CasMVS (Gu et al., 2020) aims to improve memory and time efficiency in multi-view stereo and stereo matching by introducing a hierarchical cost volume formulation. It constructs a feature pyramid to encode geometry and context, narrowing the depth or disparity range at each stage and recovering high-resolution outputs in a coarse-to-fine manner. Although CasMVS uses a similar hierarchical architecture to extract and process features, there are some core technical differences. CasMVS utilizes the depth from the previous stage as the foundation for the next stage's prediction, without incorporating mechanisms for error-aware evaluation of the coarse depth. On the other

hand, HiSplat employs the Error Aware Module and Modulating Fusion Module to refine Gaussian primitives across stages, enhancing inter-stage information exchange and improving the robustness of the results. Furthermore, HiSplat's final Gaussian primitives fuse multi-scale output Gaussian primitives in a hierarchical manner, whereas CasMVS outputs the finest level of depth estimation directly, lacking a hierarchical structure.