# OpenReview forum: "HiSplat: Hierarchical 3D Gaussian Splatting for Generalizable Sparse-View Reconstruction"
_ICLR.cc/2025/Conference — ICLR 2025 Poster_

### Official Review · Reviewer_PsL3 · 2024-10-28

**Soundness:** 4
**Presentation:** 3
**Contribution:** 3
**Rating:** 6
**Confidence:** 4

**Summary:**

This paper proposes HiSplat to estimate the 3D Gaussian Splatting structure of two given images with known camera poses in a feedforward process. It aims to use multi-scale information to improve the quality and for which two novel modules are proposed. Extensive experiments on various datasets can show its SOTA performance and good generalizability, and validate the effects of each proposed component.

**Strengths:**

- The proposed Error Aware Module and Modulating Fusion Module are interesting, which can effectively filter the errors from photometric loss in a simple way.

- Experiments are extensive. Both qualitative and quantitative results show that HiSplat achieves SOTA performance compared to the compared baselines. The results are convincing.

- The paper is easy to follow.

**Weaknesses:**

- Some related methods are missed in the discussion, e.g., FreeSplat [1] and Splatt3R [2]. Especially, FreeSplat also uses a cost-volume-based structure to estimate the depth and then Gaussians with multi-scale strategy. A step further, the architecture of this work is also somehow similar to CasMVS [3] that uses a cascade structure to solve the MVS depth estimation task. The main differences lie in how the multi-scale features are used. Need more deeper discussions about these related works to show the technical differences and more valuable analysis.

- The setting of this paper seems to be fixed at using only two images. Would like to see some discussion about if it's possible to extend the method to more input views, just like what FreeSplat does. Even if it is a part of limitation, it would still be much valuable.



[1] Wang, Yunsong, et al. "FreeSplat: Generalizable 3D Gaussian Splatting Towards Free-View Synthesis of Indoor Scenes." arXiv preprint arXiv:2405.17958 (2024).

[2] Smart, Brandon, et al. "Splatt3r: Zero-shot gaussian splatting from uncalibrated image pairs." arXiv preprint arXiv:2408.13912 (2024).

[3] Gu, Xiaodong, et al. "Cascade cost volume for high-resolution multi-view stereo and stereo matching." Proceedings of the IEEE/CVF conference on computer vision and pattern recognition. 2020.

**Questions:**

See weaknesses.

---

> ### Author Response · Authors · 2024-11-22
>
> Q1: Some related methods are missed in the discussion, e.g., FreeSplat and Splatt3R. Especially, FreeSplat also uses a cost-volume-based structure to estimate the depth and then Gaussians with multi-scale strategy. A step further, the architecture of this work is also somehow similar to CasMVS that uses a cascade structure to solve the MVS depth estimation task. The main differences lie in how the multi-scale features are used. Need more deeper discussions about these related works to show the technical differences and more valuable analysis.
>
> Answer: Thanks for your supplementary. We have added comparisons and discussions related to the mentioned works in the Appendix of the revision and will incorporate them in the main text in the final version. In the following, we provide the differences between HiSplat and the mentioned methods.
>
> 1. Comparison with FreeSplat. ①The cost-volume-based structure has been proven to be a highly effective and widely accepted component in MVS methods, such as MVSNet[1], MVSformer++[2], and LANet[3]. It has also been adopted in emerging generalizable Gaussian-Splatting-based methods, such as MVSplat, Transplat, and FreeSplat. Considering the high effectiveness of cost-volume-based structure in geometry feature extraction and depth estimation, we follow them to introduce this structure for a more accurate location of Gaussian primitives. It is worth noting that it is not claimed as a novelty in this manuscript. ②Additionally, FreeSplat employs a multi-scale U-Net as an efficient feature aggregator. However, the features it ultimately uses to predict Gaussian attributes are from the last layer and still single-scale, resulting in single-scale Gaussian primitives without hierarchical structure. On the contrary, HiSplat leverages multi-scale features from different layers of the U-Net to generate different-scale Gaussians, which represent the scene from bone to flesh.
>
> 2. Comparison with Splatt3R. Splatt3R focuses more on pose-free novel view synthesis, while HiSplat primarily emphasizes using hierarchical Gaussian primitives for better scene representation. These are two orthogonal directions. We believe extending HiSplat in pose-free settings is a promising future direction.
>
> 3. Comparison with CasMVS. ① Different tasks. It is worth noting that CasMVS and HiSplat address different tasks within distinct subfields. CasMVS focuses on multi-view depth estimation, while HiSplat focuses on novel sparse-view synthesis. Although both methods employ coarse-to-fine architectures, experiments in Table 3 demonstrate that directly leveraging multi-scale features to construct hierarchical Gaussian representation yields suboptimal performance. The core effectiveness of HiSplat lies in its Error Aware Module (EAM) and Modulating Fusion Module (MFM). ② Different purposes. The primary objective of CasMVS is to reduce the depth candidates in the cost volume via coarse-to-fine depth prediction, thereby minimizing memory and computational overhead. In contrast, HiSplat aims to generate hierarchical Gaussian primitives for better 3D scene representation, which may slightly increase computational cost. ③ Different techniques. CasMVS utilizes the depth from the previous stage as the foundation for the next stage's prediction, without incorporating error-aware evaluation of the coarse depth. On the other hand, HiSplat employs the Error Aware Module and Modulating Fusion Module to refine Gaussian primitives across stages, enhancing inter-stage information exchange and improving the robustness of the results. Furthermore, HiSplat's final Gaussian primitives fuse multi-scale output Gaussian primitives in a hierarchical manner, whereas CasMVS outputs the finest level of depth estimation directly, lacking a hierarchical structure.
>
> Q2: The setting of this paper seems to be fixed at using only two images. Would like to see some discussion about if it's possible to extend the method to more input views, just like what FreeSplat does. Even if it is a part of limitation, it would still be much valuable.
>
> Answer: HiSplat is easy to extend to reconstruct from multiple images, Please see common question 2.
>
> [1] Mvsnet: Depth inference for unstructured multi-view stereo.
>
> [2] MVSFormer++: Revealing the Devil in Transformer's Details for Multi-View Stereo.
>
> [3] Long-range Attention Network for Multi-View Stereo.

---

> > ### Comment · Reviewer_PsL3 · 2024-11-23
> >
> > Thanks for the reply. My main concerns have been well solved. Nevertheless, considering the aggregation of multi-scale features has already been considered by FreeSplat months ago, maybe the authors should revisit if it's sound to declare HiSplat is the first to study and introduce hierarchical 3D Gaussians representation in generative 3DGSs. Especially, the words "hierarchical 3D Gaussians representation" may be ambiguous and disputed here, as it likely refers to methods with LOD such as [1] [2]. Overall, I vote for acceptance.
> >
> > [1] A *Hierarchical 3D Gaussian Representation* for Real-Time Rendering of Very Large Datasets, SIGGRAPH 2024
> >
> > [2] Octree-GS: Towards Consistent Real-time Rendering with LOD-Structured 3D Gaussians

---

> ### Author Response · Authors · 2024-11-24
>
> Thanks for your approval and timely response! We will clarify in the final version that FreeSplat has used multi-scale features in generalizable 3D-GS, and we will add a footnote specifically stating that the hierarchical 3D Gaussian representation in the context of our work refers to using hierarchical Gaussian primitives to represent the scene, rather than using multi-scale features to generate Gaussians, avoiding potentially overstating our contribution. Additionally, we will include a clarification in the final version to distinguish our hierarchical 3D Gaussian representation from similar concepts in LOD methods to avoid ambiguity and controversy. Specifically, LOD-based methods such as [1] [2] typically build a hierarchical Gaussian tree for the scene, where Gaussians with different granularities are switched and selected dynamically during rendering based on spatial location, improving rendering speed and reducing rendering overhead, while maintaining quality. In contrast, our method focuses on high performance, and instead of switching or selecting Gaussians, we use hierarchical Gaussians simultaneously to represent both small-scale texture details and large-scale structures in the scene, which slightly increases rendering overhead.
>
> [1] A Hierarchical 3D Gaussian Representation for Real-Time Rendering of Very Large Datasets, SIGGRAPH 2024
>
> [2] Octree-GS: Towards Consistent Real-time Rendering with LOD-Structured 3D Gaussians

---

### Official Review · Reviewer_R9EA · 2024-11-02

**Soundness:** 3
**Presentation:** 4
**Contribution:** 3
**Rating:** 6
**Confidence:** 4

**Summary:**

This work presents a new 3DGS-based generalizable sparse-view scene reconstruction method, HiSplat, which effectively integrates multi-scale hierarchical MVS features with proposed EAM and MFM modules to enhance the quality of feed-forward 3DGS models, especially in fine-grained details. Experimental results demonstrate the superior reconstruction quality compared to prior SoTA methods.

**Strengths:**

1. The paper is well-motivated from the prior works on multi-scale hierarchical visual knowledge. It is well-written.
2. The way the authors integrate the multi-scale features is clever and interesting. I love the Error Aware part, HiSplat wisely incorporates the reconstruction error refinement, usually through gradient optimizations, into this feed-forward reconstruction model.  The error maps are used to guide the refinement on the higher levels. I believe this idea can also be beneficial for other related tasks.
3. The analysis in 4.4 is very informative and useful. The figures effectively demonstrate how multi-scale features can help improve the reconstruction quality.

**Weaknesses:**

1. The current model is pretty complex. It contains a lot of different neural models (MVSformer++, DINOv2, and a bundle of UNets and MLPs). I am not very clear which modules are trainable and which are not. The authors should clarify and summarize this in 3.6. Also, it would be even better, this is not required for rebuttal, if HiSplat could be a relatively simpler unified foundation model (e.g., like LRM).
2. Although the writing is generally good, the citation format of this paper is a bit disturbing. Please properly use `\cite` and `\citep` provided in the ICLR template.
3. Minor typos:
    * MVSpalt → MVSplat
    * L298: fellow → follow

**Questions:**

1. Since it is unlikely we can always get accurate camera poses from sparse views in real life, can this work be extended to handle the pose-free sparse-view reconstruction for the captures from real life? If it is easy to implement, it would be great to see some examples.
2. The depth coefficient $\eta$ is a bit tricky. I assume it is a fixed value for each stage after the training, is this correct? If so, how do you choose the proper $\eta$ values?

---

> ### Author Response · Authors · 2024-11-22
>
> Q1: The current model is pretty complex. It contains a lot of different neural models (MVSformer++, DINOv2, and a bundle of UNets and MLPs). I am not very clear which modules are trainable and which are not. The authors should clarify and summarize this in 3.6. Also, it would be even better, this is not required for rebuttal, if HiSplat could be a relatively simpler unified foundation model (e.g., like LRM).
>
> Answer: We sincerely apologize for not providing detailed clarification on this part. We have added this in Section 3.6 in the updated revision, explaining which components require training. Specifically, only the feature extractor from DINOv2 is frozen, while all other components of HiSplat remain trainable. Besides, simplifying HiSplat is an exciting direction for future work, and we look forward to making further improvements to HiSplat. For instance, replacing the CNN modules with a fully transformer-based framework can simplify the architecture, enabling it to serve as a more straightforward foundation model.
>
> Q2: Although the writing is generally good, the citation format of this paper is a bit disturbing. Please properly use \cite and \citep provided in the ICLR template. And there are some minor typos.
>
> Answer: Thanks for the suggestion! We have corrected it in the updated revision.
>
> Q3: Since it is unlikely we can always get accurate camera poses from sparse views in real life, can this work be extended to handle the pose-free sparse-view reconstruction for the captures from real life? If it is easy to implement, it would be great to see some examples.
>
> Answer: HiSplat adopts the mainstream settings that require the calibrated camera parameters for sparse-view reconstruction, following methods like MVSPlat and PixelSplat, without delving deeply into pose-free settings. However, with the development of MVS methods, such as DUSt3R[1], enable the fast and simple extraction of camera parameters from sparse-view images, and the whole procedure consumes less than 1 second for two images. By integrating DUSt3R, HiSplat can easily perform sparse-view reconstruction without relying on pre-calibrated camera parameters.
>
> We conduct a simple experimental setup to test this on the DTU and Replica datasets, using sparse views without camera parameters. Specifically, we use pre-trained DUSt3R to extract camera parameters, including intrinsics and extrinsics, from input images and then directly apply HiSplat (pre-trained on RealEstate10K) for novel view synthesis. The results are shown in Table t3. Compared with using relatively accurate poses from the dataset, using poses generated by DUSt3R leads to reduced performance but still outperforms MVSplat. This is because the simple experimental setup lacks sufficient coupling between DUSt3R’s pose extraction and HiSplat’s reconstruction process. The resulting less accurate poses affect HiSplat's ability to precisely determine the position and shape of Gaussian primitives, thus leading to reduced performance.
>
> Additionally, upon the reviewer’s request, we test a real-life scenario using three casually taken photos with a smartphone: two as reference images and one as the ground truth. Using the same simplified pipeline described above, the results are shown in Fig. 10 in the revision. It demonstrates that HiSplat+DUSt3R's simplified pipeline has the ability to handle sparse-view reconstructions in real-life settings where only the images are inputted, producing a slightly satisfactory novel view synthesis. We believe designing a more robust and systematic pose-free reconstruction method based on HiSplat is a promising future direction.
>
> Q4: The depth coefficient is a bit tricky. I assume it is a fixed value for each stage after the training, is this correct? If so, how do you choose the proper values?
>
> Answer: The depth coefficient is a manually set constant determined before training and remains fixed throughout training and inference. As a hyperparameter, it only requires a relatively small value, such as 0.1, to ensure that the depth difference between the decorative Gaussian primitives and the skeletal Gaussian primitives does not become excessively large.
>
> **Table t3**: Cross-dataset evaluation with pose-free settings. HiSplat+GT pose: using the camera parameters from datasets; HiSplat+DUSt3R pose: using the camera parameters generated by pre-trained DUSt3R, which is used for pose-free settings. By simply combining with DUSt3R, HiSplat can handle pose-free sparse-view reconstruction.
> |        Method       | RealEstate10K→DTU | RealEstate10K→Replica |
> |:-------------------:|:-----------------:|:---------------------:|
> |   MVSplat+GT pose   |       13.94       |         23.79         |
> | MVSplat+DUSt3R pose |       13.18       |         21.04         |
> |   HiSplat+GT pose   |       16.05       |         27.21         |
> | HiSplat+DUSt3R pose |       14.58       |         22.20         |
>
> [1] DUSt3R: Geometric 3D Vision Made Easy.

---

> ### Author Response · Authors · 2024-11-24
> **Follow-up on the response**
>
> Dear reviewer,
>
> We hope our response has addressed your questions and concerns. If so, could we kindly ask you to consider increasing the score? Thank you once again for your valuable and insightful feedback!

---

> ### Comment · Reviewer_R9EA · 2024-11-24
>
> Thanks for the author's detailed responses. The revised paper now looks good to me. All my questions are properly addressed.
>
> I will keep my positive rating score for this submission.

---

### Official Review · Reviewer_gQRm · 2024-11-02

**Soundness:** 2
**Presentation:** 3
**Contribution:** 2
**Rating:** 6
**Confidence:** 4

**Summary:**

This paper claims that previous generalizable 3D Gaussian Splatting methods utilize uniform 3D Gaussians, making it challenging to simultaneously capture large-scale structures and intricate texture details. The authors propose a hierarchical approach to generalizable 3D Gaussian Splatting, constructing hierarchical 3D Gaussians through a coarse-to-fine strategy. Additionally, they design an Error Aware Module and a Modulating Fusion Module to manage interactions among different hierarchical Gaussians.

**Strengths:**

1. The idea of using different hierarchical Gaussians to simultaneously capture large-scale structures and delicate textures in scene generation sounds reasonable.
2. The ablative study seems complete.
3. The motivation is clear since not all points in a single image are equally important.

**Weaknesses:**

1. The paper introduces many modules, which slows down the process compared to MVSplat, which is also revealed in Tab.4, after adding all the stages together.
2. The improvement in metrics compared with baseline method does not seem very significant from the reviewer's perspective. But the generalization capability to unseen datasets like DTU seems interesting. The authors can show more cases that demonstrates generalization capability.
3. Some of the visualized cases does not seem to possess a very significant improvement compared with MVSplat. For example, the last line of Fig.6; the left red bounding box in the first example in Fig. 1.

**Questions:**

Apart from the weakness part, I also have the following questions.
1. Will the number of Gaussian primitives change significantly after applying this hierarchical structure? A comparison with baseline methods on the primitive numbers will be nice.
2. It seems that the first example in Fig.6. shows that HiSplat can generate less artifacts in the occluded area, as highlighted by the red bounding box. Does this just happen in coincidence or is it a prevalent phenomenon?

I still have the aforementioned concerns and I would be glad to raise my rating if they can be addressed.

---

> ### Author Response · Authors · 2024-11-22
>
> Q1: The paper introduces many modules, which slows down the process compared to MVSplat, which is also revealed in Tab.4, after adding all the stages together.
>
> Answer: Please see common question 1.
>
> Q2: The improvement in metrics compared with baseline method does not seem very significant from the reviewer's perspective. But the generalization capability to unseen datasets like DTU seems interesting. The authors can show more cases that demonstrates generalization capability.
>
> Answer:
> 1. Since TranSplat has not been open-sourced and is still in preprint, nearly contemporaneous with our work, we use MVSplat as the primary baseline for comparison in this manuscript. Compared with prior methods quantitatively, our performance improvements are relatively significant. For instance, while MVSplat (ECCV2024 Oral) improves +0.5 PSNR on RealEstate10K and +0.11 PSNR on ACID compared with its baseline PixelSplat, HiSplat further surpasses MVSplat, achieving additional gains of +0.82 PSNR on RealEstate10K and +0.5 PSNR on ACID.
>
> 2. In addition to Fig. 3 and Fig. 7, we randomly select and provide more cases to demonstrate the cross-dataset generalization of HiSplat on DTU in Fig. 11. It can be observed that compared with other methods, HiSplat can consistently obtain high-quality rendering images from novel views in the out-of-distribution scenes.
>
> Q3: Some of the visualized cases does not seem to possess a very significant improvement compared with MVSplat. For example, the last line of Fig.6; the left red bounding box in the first example in Fig. 1.
>
> Answer: We apologize for not clearly explaining the differences of cases in Fig. 6 and Fig. 1. In the final version, we will add detailed explanations in the captions of Fig. 6 and Fig. 1.
>
> Here are the detailed differences.
>
> In Fig. 6, the lumination of the clothes on the bed rendered by MVSplat is wrong, while HiSplat can recover the right color and lighting. In Fig. 1, compared with MVSplat, HiSplat can achieve more accurate localization of objects, such as the relative position between the cabinet and the hanging items behind it shown in the left red box, and fewer artefacts shown in the right red box.
>
> Q4: Will the number of Gaussian primitives change significantly after applying this hierarchical structure? A comparison with baseline methods on the primitive numbers will be nice.
>
> Answer:  Thanks for the suggestion. We have added the comparison with baseline methods on the primitive numbers in Table 7 in the revision. In the third stage, the number of Gaussian primitives is the same as in MVSPlat. However, as the stage index is reduced, the number of Gaussian primitives decreases exponentially by a factor of 4, resulting in significantly fewer Gaussian primitives in the early stages. Therefore, the total number of Gaussian primitives is only 1.3125 (0.0625 + 0.25 + 1) times compared with MVSPlat. The comparison of the number of Gaussian primitives and performance is shown in Table t2. Compared to other methods, HiSplat achieves significant performance improvement with only a slight increase in Gaussian primitives. It is worth noting that even if MVSplat triples the number of Gaussian primitives, it does not yield noticeable improvement, indicating that the increase in Gaussian primitives is not the main reason for HiSplat's performance gain.
>
> Q5:It seems that the first example in Fig.6. shows that HiSplat can generate less artifacts in the occluded area, as highlighted by the red bounding box. Does this just happen in coincidence or is it a prevalent phenomenon?
>
> Answer: This is not a coincidental phenomenon, and it can also be observed in the two example images in Fig. 1. This advantage originates from:
>
> 1. Error Aware Module: During network inference, if artifacts occur in the early stages, they inevitably lead to significant errors. HiSplat can perceive these errors through the error-aware module and correct them in the subsequent stages.
>
> 2. Hierarchical Gaussian representation. The hierarchical Gaussian primitives provide stronger representation ability. For one area, HiSplat leverages different-scale Gaussian primitives for representation. The more sophisticated scene representation offers extrapolation ability, particularly for occluded areas. When processing occluded areas, the large-scale Gaussian primitives within the hierarchical Gaussian representation can extend across wider ranges, enabling extrapolated rendering to roughly compensate for missing information, which prevents noticeable cracks and artefacts. In contrast, other methods that only rely on single-scale small Gaussian primitives lack this capability, resulting in more remarkable artefacts.

---

> > ### Comment · Reviewer_gQRm · 2024-11-25
> >
> > Thank you for your detailed reply! I still have a minor concern regarding the original Question 3:
> >
> > If I understand correctly, the clothes in Figure 6 should appear completely dark under the novel view, and the plate in Figure 1 should be positioned higher. Despite these observations, the PixelSplat results seem satisfactory in both examples. I suggest including some more pronounced comparisons in the final version.
> >
> > But overall, I am willing to raise my rating.

---

> > > ### Author Response · Authors · 2024-11-25
> > >
> > > Thanks for your support and approval!  We sincerely appreciate your constructive feedback and valuable advice! We apologize that the LPIPS of MVSplat (1 Gaussian per pixel) is miscopied, and we have corrected it. Following your advice, we will provide more convincing comparisons in the final version.

---

> ### Author Response · Authors · 2024-11-22
>
> **Table t2**: Comparison of different Gaussian primitive number. Compared with other pixel-wiseGaussian-based methods, HiSplat utilizes hierarchical Gaussian representation, which increases marginally the number of Gaussian primitives and boosts the performance to achieve a better tradeoff.
> |             Method            | Gaussian Primitive Number | PSNR↑ | SSIM↑ | LPIPS↓ |
> |:-----------------------------:|:-------------------------:|:-----:|:-----:|:------:|
> |           PixelSplat          |        65,632(×1.0)       | 25.89 | 0.858 |  0.142 |
> | MVSplat(1 Gaussian per pixel) |        65,632(×1.0)       | 26.39 | 0.869 |  0.128 |
> | MVSplat(3 Gaussian per pixel) |       196,896(×3.0)       | 26.54 | 0.872 |  0.127 |
> |            HiSplat            |      86,142(×1.3125)      | 27.21 | 0.881 |  0.117 |

---

> ### Author Response · Authors · 2024-11-24
> **Follow-up on the response**
>
> Dear reviewer,
>
> We hope our response has addressed your questions and concerns. If so, could we kindly ask you to consider increasing the score? Thank you once again for your valuable and insightful feedback!

---

### Official Review · Reviewer_Lyvm · 2024-11-02

**Soundness:** 3
**Presentation:** 3
**Contribution:** 3
**Rating:** 6
**Confidence:** 4

**Summary:**

HiSplat introduces a coarse-to-fine strategy to construct hierarchical 3D Gaussians for generalizable 3D Gaussian Splatting. Additionally, HiSplat proposes an Error-Aware Module and a Modulating Fusion Module to enhance inter-scale interactions. Comprehensive experiments demonstrate that HiSplat achieves better results than single-scale methods.

**Strengths:**

1. HiSplat first introduces the hierarchical 3D Gaussian representation in the task of generalizable 3D Gaussian Splatting.
2. The experiment effectively demonstrated the effectiveness of its hierarchical representation through Fig 4, which shows the primitives of Gaussians obtained from different stages.
3. This paper show the effective ablation study, comparing the impact of different modules on the performance of Generalizable Sparse-View Reconstruction.

**Weaknesses:**

1. Several spelling errors have been identified in the document, specifically on lines 73, 106, and 508.
2. The hierarchical representations appear to be memory-intensive and computationally inefficient. It is recommended that the paper discusses the implications for memory, FLOPs, and inference time.
3. The rationale for incorporating the DINO feature seems underwhelming, and the provided experiments do not convincingly demonstrate the necessity of including the DINO feature.
4. I hope to see additional experiments conducted to explore the impact of increasing the number of perspectives on the quality of reconstruction. It appears that the majority of the experiments presented in the article are limited to just two perspectives, which may not fully capture the potential benefits of a multi-perspective approach.

**Questions:**

1. Discuss more about the importance of incorporating the DINO feature.
2. Discuss more about the complexity analysis of the method.

---

> ### Author Response · Authors · 2024-11-22
>
> Q1: Several spelling errors have been identified in the document, specifically on lines 73, 106, and 508.
>
> Answer: Thanks for your advice! We have corrected it in the revision.
>
> Q2: The hierarchical representations appear to be memory-intensive and computationally inefficient. It is recommended that the paper discusses the implications for memory, FLOPs, and inference time. (Discuss more about the complexity analysis of the method.)
>
> Answer: Please see common question 1.
>
> Q3: The rationale for incorporating the DINO feature seems underwhelming, and the provided experiments do not convincingly demonstrate the necessity of including the DINO feature. (Discuss more about the importance of incorporating the DINO feature.)
>
> Answer: Our introduction of DINO features is inspired by MVSformer++. When building the framework of HiSplat, we introduced the DINO feature experimentally and results show that, while it introduces almost no additional trainable parameters, it does improve the network's reconstruction capability (+0.19 PSNR in RealEstate10K) and generalization performance (+0.1 PSNR in DTU). Thus, we keep it in our framework. However, it is not part of the claimed novelty and contributions. We consider the DINO feature as an enhancement technique. In fact, as shown in Table 3, the performance gain from DINO features is relatively lower compared with our proposed Error Aware Module and Modulating Fusion Module. Therefore, for scenarios where inference or training efficiency is critical, the DINO feature related components can be eliminated.
>
>
> Q4: I hope to see additional experiments conducted to explore the impact of increasing the number of perspectives on the quality of reconstruction. It appears that the majority of the experiments presented in the article are limited to just two perspectives, which may not fully capture the potential benefits of a multi-perspective approach.
>
> Answer: Please see common question 2.

---

> ### Author Response · Authors · 2024-11-24
> **Follow-up on the response**
>
> Dear reviewer,
>
> We hope our response has addressed your questions and concerns. If so, could we kindly ask you to consider increasing the score? Thank you once again for your valuable and insightful feedback!

---

> ### Author Response · Authors · 2024-11-25
>
> Dear reviewer,
>
> We apologize for disturbing you again. As the discussion phase is nearing its end, we would be grateful to hear your feedback and wondered if you might still have any concerns we could address. We hope our reply could address your questions.  It would be appreciated if you could raise your score on our paper. We thank you again for your effort in reviewing our paper.

---

> > ### Comment · Reviewer_Lyvm · 2024-11-25
> >
> > I appreciate the author's detailed responses. They have effectively addressed my concerns, and I will maintain my positive rating.

---

### Author Response · Authors · 2024-11-22
**The response to all reviewers**

We appreciate the reviewers (R1:Lyvm, R2:gQRm, R3:R9EA, R4:PsL3) for their constructive suggestions and approving our contributions:
(1) interesting, well-motivated and novel approach (R1, R2, R3, R4); (2) comprehensive and complete experiments (R1, R2, R4); (3) well-writen and easy to follow (R3, R4). (4) informative and useful analysis (R1, R3).

We have updated the revision to follow the reviewers' suggestions, including correcting some errors, adding some experimental results, supplying discussion, and declaring confusion. To indicate the corresponding changes suggested by reviewers, we use a unique text color for each reviewer(R1:red, R2:blue, R3:green, R4:cyan) in the updated revision.  We will gladly incorporate all the feedback in the final version under the ICLR policy. We provide unique responses to each reviewer’s specific questions individually. Here, we respond to some common questions.

---

### Author Response · Authors · 2024-11-22

Common Question 1(R1Q2, R2Q1): HiSplat introduces many modules slowing down the process. Discuss the computational and memory costs of the method in detail.

Answer: We have discussed the cost of memory and computation in Appendix A.2. Besides, we also provide more details and analysis as follows.
1. Sources of additional inference overhead.  The extra overhead is from ① more Gaussian primitives: Compared with previous pixel-wise Gaussian methods such as MVSplat and PixelSplat, HiSplat introduces three-stage hierarchical Gaussian representation, resulting in approximately 1.31x Gaussian primitives. However, thanks to the highly efficient parallelizable rasterization library of Gaussian Splatting, the inference time during rendering only marginally increases (approximately 1.1x). ②more computational modules: we incorporate a pre-trained DINOv2 feature extractor based on a ViT-B/14 architecture. Moreover, we propose Error Aware Module and Modulating Fusion Module, mainly composed of lightweight U-Nets and MLPs. These introduced modules increase the computational demands of inference, which account for the majority of the inference time. Nevertheless, as shown in Table 3 in the manuscript, they play an important role in processing hierarchical Gaussian representation, delivering higher novel view synthesis performance and improved generalization. Thus, we believe the additional inference overhead is valuable. It is hard to have a free lunch. Besides, the total computational cost of our full-version approach is higher than MVSplat (about 2x) but lower than PixelSplat, indicating that the efficiency remains within an acceptable range.
2. Potential techniques for cost reduction. The current HiSplat reported in the manuscript is an experimental version, focusing on evaluating performance and feasibility rather than efficiency. There are several potential strategies to reduce inference costs in practical deployment: ①Pipeline parallelism. Because HiSplat can be divided into three stages after extracting features using the U-net backbone, pipeline parallelism can be employed for multiple input batches easily. Utilizing existing pipeline parallelism libraries can further enhance HiSplat's efficiency. ② Dynamic network manner. Hisplat's unique hierarchical design supports dynamic networks for flexible implementation. When efficiency is required, as shown in Table 4 in the manuscript, it is possible to use only the Gaussians in the early stages to reconstruct the scene, which sacrifices the performance but reduces inference time and memory requirements. HiSplat offers more deployment flexibility compared to methods with fixed-resolution Gaussians. ③Transformer acceleration techniques. A significant portion of computational and memory overhead in HiSplat arises from transformer-based modules like the corresponding modules of DINOv2 and UniMatch. Techniques like flash attention[1] and other transformer-specific optimizations can substantially reduce memory and computation requirements without sacrificing performance.

---

### Author Response · Authors · 2024-11-22

Common Question 2(R1Q4, R4Q2): Can HiSplat be extended to more input views? More experiments are needed.

Answer: Although HiSplat focuses on two-view reconstruction, the main components of HiSplat, such as the hierarchical Gaussian representation, shared feature extractors, Error Aware Module, and Modulating Fusion Module, are view-invariant, which means that HiSplat can be easily extended to handle reconstruction from an arbitrary number of views. To perform an initial verification, we directly use the weights trained by inputting two views on RealEstate10K to conduct cross-dataset testing with three views on DTU and Replica. The results are shown in Table t1. Compared with the two-view cases, the performance slightly improves with inputting three views, and HiSplat still outperforms MVSplat, demonstrating the feasibility of using HiSplat for multi-view reconstruction directly. However, since HiSplat is not specifically trained for three-view tasks and lacks specific structural adjustments for more views in Table t1, the gain is marginal. Considering the limited experimental time during the rebuttal phase, we will provide additional results on training and testing under more multi-view settings in the final version. Additionally, the core contribution of HiSplat, i.e., the hierarchical Gaussian representation, has the potential to be integrated with recent multi-view reconstruction methods, such as FreeSplat, to enhance their ability of novel view synthesis.

**Table t1**: Cross-dataset evaluation with more reference views. We train the models on RealEstate10K with two reference views and test them on DTU and Replica with three reference views.
| Method          | RealEstate10K→DTU | RealEstate10K→Replica |
|-----------------|-------------------|-----------------------|
| MVSplat+2 views | 13.94             | 23.79                 |
| MVSplat+3 views | 14.29             | 24.75                 |
| HiSplat+2 views | 16.05             | 27.17                 |
| HiSplat+3 views | 16.35             | 27.47                 |


[1] Flashattention-2: Faster attention with better parallelism and work partitioning.

---

### Meta-Review · Area_Chair_JFST · 2024-12-18

**Metareview:**

This paper introduces a coarse-to-fine strategy to construct hierarchical 3D Gaussians for generalizable sparse-view 3D Gaussian Splatting. In addition, this paper proposes an Error-Aware Module and a Modulating Fusion Module to promote inter-scale interactions. Comprehensive experiments demonstrate that the proposed method achieves better results than single-scale methods.

Main strengths of this paper are as below:
- The paper is clearly written and easy to follow.
- The paper is well-motivated from the prior works on multi-scale hierarchical visual knowledge.
- The paper introduces a coarse-to-fine strategy into sparse-view Gaussian Splatting. And it proposes an Error-Aware Module and a Modulating Fusion Module to promote inter-scale interactions, which are innovative and inspiring.
- The experiments demonstrate improvements over previous methods. And the ablation study shows the effectiveness of Error Aware Module and Modulating Fusion Module.

Needing more experiments and analyses were raised by the reviewers. Please revise the paper according to the discussions before submitting the final version.

**Additional Comments On Reviewer Discussion:**

- Reviewer Lyvm suggested that the authors further discuss the impact of hierarchical representations on memory, FLOPs, and inference time. And Reviewer Lyvm advised analyzing the benefits of incorporating DINO features. The authors provided experimental results to analyze and explain the issues mentioned above.
- Reviewer gQRm suggested that the authors show more examples of generalization. The authors provided more examples and analysis to illustrate the generalizability of the proposed method.
- Reviewer R9EA asked whether the proposed method can be extended to handle the pose-free sparse-view reconstruction for the captures from real life. The authors responded that the proposed method can easily perform sparse-view reconstruction without relying on pre-calibrated camera parameters by integrating MVS methods such as DUSt3R.
- Reviewer PsL3 raised that some related methods are missed in the discussion, e.g., FreeSplat and Splatt3R. The authors presented a detailed analysis comparing these methods with the proposed method and explained the differences between the methods.

---

### Decision · Program_Chairs · 2025-01-22

Accept (Poster)